# RETHINKING MULTI-DOMAIN GENERALIZATION WITH A GENERAL LEARNING OBJECTIVE

## ABSTRACT

Multi-domain generalization (mDG) is universally aimed at diminishing the gap between training and testing distribution, which in turn facilitates the learning of a mapping from marginal distributions to labels. However, in the literature of mDG, a general learning objective paradigm is conspicuously missing, and the constraint of a static target's marginal distribution is often present. In this paper, we propose to leverage a $\mathbf{Y}$-mapping $\psi$ to relax the constraint. We then rethink the learning objective for mDG and design a new **general learning objective** that can be used to interpret and analyze most existing mDG wisdom. This general objective is bifurcated into two synergistic amis: learning domain-independent conditional features, and maximizing a posterior. Explorations also extend to two effective regularization terms that incorporate prior information and suppress invalid causality, alleviating the issues that come with relaxed constraints. Inspired by the Generalized Jensen-Shannon Divergence, we contribute to deriving an upper bound for the domain alignment of domain-independent conditional features, disclosing that many previous mDG endeavors actually **optimize partially the objective** and thus lead to limited performance. As such, the general learning objective is simplified into four practical components and can be easily used in various tasks and different frameworks. Overall, our study proposes a general, robust, and flexible mechanism to handle complex domain shifts. Extensive empirical results indicate that the proposed objective with $\mathbf{Y}$-mapping leads to substantially better mDG performance.

## 1 INTRODUCTION

Domain shift, which breaks the independent and identical distributed (*i.i.d.*) assumption amid training and test distributions (Wang et al., 2022), poses a common yet challenging problem in real-world scenarios. Multi-domain generalization (mDG) (Blanchard et al., 2011)) is garnering increasing attention owing to its promising capacity to utilize multiple distinct but related source domains for model optimization, ultimately intending to generalize well to unseen domains. Intrinsically, the primary objective for mDG is the maximization of the joint distribution between observations $\mathbf{X}$ and targets $\mathbf{Y}$ across all domains $\mathcal{D}$:

$$\max P(\mathbf{X}, \mathbf{Y} \mid \mathcal{D}) = P(\mathbf{Y} \mid \mathcal{D})P(\mathbf{X} \mid \mathbf{Y}, \mathcal{D}) = P(\mathbf{X} \mid \mathcal{D})P(\mathbf{Y} \mid \mathbf{X}, \mathcal{D}). \tag{1}$$

A prevalent approach initiates by maximizing the marginal distribution $P(\mathbf{X}|\mathcal{D})$ before presuming an invariant $P(\mathbf{Y}|\mathbf{X}) = P(\mathbf{Y}|\mathbf{X}, \mathcal{D})$ across domains (Zhou et al., 2022), anchored on an assumption that $P(\mathbf{Y}|\mathcal{D})$ remains consistency across domains $\mathcal{D}$.

*Is $P(\mathbf{Y}|\mathcal{D})$ truly static across domains?* In other words, does $\mathbf{Y}$ truly lack domain-dependent features? In classification tasks, typically, the influence of $\mathcal{D}$ on $\mathbf{Y}$ is substantially marginal. However, this assumption is not universally applicable, particularly in tasks such as regression or segmentation. Subsequently, MDA (Hu et al., 2020) relaxes the assumption of stable $P(\mathbf{Y}|\mathcal{D})$ by providing an average class discrepancy, allowing both $P(\mathbf{X}|\mathbf{Y}, \mathcal{D})$ and $P(\mathbf{Y}|\mathcal{D})$ vary across $\mathcal{D}$. However, MDA has to conduct class-specific sample selection under domains for obtaining $P(\mathbf{X}|\mathbf{Y}, \mathcal{D})$, which constrains its objective's universality and struggles with tasks beyond basic classification especially where $\mathbf{Y}$ is not discrete. To better tackle the $\mathcal{D}$-dependent variations in both $\mathbf{X}$ and $\mathbf{Y}$, we introduce two learnable mappings, $\phi$ and $\psi$, that project $\mathbf{X}$ and $\mathbf{Y}$ into the **same** latent Reproducing Kernel

Table 1: A summary of objectives of ERM (Gulrajani & Lopez-Paz, 2020), DANN (Ganin et al., 2016), CORAL (Sun & Saenko, 2016), CDANN Li et al. (2018d), CIDG (Li et al., 2018c), MDA (Hu et al., 2020), MIRO (Junbum et al., 2022), SIMPLE (Li et al., 2022), RobustNet (Choi et al., 2021) and our approach. All constants are omitted here. 'Others' denote other methods that are not specified in the group. See more mathematical details in Appendix A.

| | **Aim1: Learning domain invariance** | **Reg1: Integrating prior** |
|---|---|---|
| ERM,CORAL,RobustNet | None | None |
| DANN | $\min_{\phi} H(P(\phi(\mathbf{X}) \mid \mathcal{D}))$ | None |
| CDANN, CIDG, MDA | $\min_{\phi} H(P(\phi(\mathbf{X}), \mathbf{Y} \mid \mathcal{D}))$ | None |
| MIRO, SIMPLE | None | $\min_{\phi} D_{\mathrm{KL}}(P(\phi(\mathbf{X}), \mathbf{Y})\|\mathcal{O})$ |
| **Ours** | $\min_{\phi,\psi} H(P(\phi(\mathbf{X}), \psi(\mathbf{Y}) \mid \mathcal{D}))$ | $\min_{\phi,\psi} D_{\mathrm{KL}}(P(\phi(\mathbf{X}), \mathbf{Y})\|\mathcal{O})$ |
| | **Aim2: Maximizing A Posterior (MAP)** | **Reg2: Suppressing invalid causality** |
| Others | $\min_{\phi} H(P(\mathbf{Y}, \phi(\mathbf{X})))$ | None |
| CORAL | $\min_{\phi} H(P(\mathbf{Y}, \phi(\mathbf{X})))$ | $\min_{\phi} -H(P(\phi(\mathbf{X}, \mathcal{D}))) + H(P(\phi(\mathbf{X})))$ |
| MDA,RobustNet | $\min_{\phi} H(P(\mathbf{Y}, \phi(\mathbf{X})))$ | $\min_{\phi} -H(P(\phi(\mathbf{X}) \mid \mathbf{Y})) + H(P(\phi(\mathbf{X})))$ |
| **Ours** | $\min_{\phi,\psi} H(P(\mathbf{Y}, \phi(\mathbf{X}))) + H(P(\mathbf{Y}, \psi(\mathbf{Y})))$ | $\min_{\phi} -H(P(\phi(\mathbf{X}) \mid \psi(\mathbf{Y}))) + H(P(\phi(\mathbf{X})))$ |

Table 2: A summary of notations.

| Symbols | Descriptions |
|---|---|
| $d_n \in \mathcal{D}, n \le N ; d' \in \mathcal{D}.$ | The $n$-th observed domains in all domains; Unseen domains in all domains. |
| $\mathbf{X}, \mathbf{Y}; \mathbf{X}_n, \overline{\mathbf{Y}}_n; \mathbf{X}', \mathbf{Y}'.$ | All observations and targets; Observations and targets in $d_n$; Observations and targets in $d'$. |
| $P(x).$ | Distributions where $x$ correspond to the random variables. |
| $\phi, \psi.$ | Learnable transformations that codify $\mathbf{X}, \mathbf{Y}$ into the same latent RKHS. |
| $\phi(\mathbf{X}), \psi(\mathbf{Y}).$ | Mapped $\mathbf{X}, \mathbf{Y}$. Within the RKHS realm, $\phi(\mathbf{X}), \psi(\mathbf{Y})$ follow Multivariate Gaussian Distributions. |
| $\mathcal{O}; R(\cdot); \sigma_{\cdot,\cdot}.$ | Prior knowledge (oracle model); Empirical risks; Covariance between two variables. |
| $\mathcal{C} : \phi(\mathbf{X}), \psi(\mathbf{Y}) \to \mathbf{Y}.$ | Predictor that predicts $\mathbf{Y}$ from $\phi(\mathbf{X}), \psi(\mathbf{Y})$. |
| $D_{\mathrm{KL}}(\cdot\|\cdot); H(\cdot); H_c(\cdot); I(\cdot;\cdot).$ | KL divergence; Entropy; Cross-entropy; Mutual information. |

Hilbert Space (RKHS), assumed to extract $\mathcal{D}$-independent features from $\mathbf{X}, \mathbf{Y}$. Incorporating these, Equation 1 can be changed as

$$\max_{\phi,\psi} P(\phi(\mathbf{X}), \psi(\mathbf{Y})), \quad \text{s.t.,} \quad \phi(\mathbf{X}), \psi(\mathbf{Y}) \perp\!\!\!\perp \mathcal{D}. \tag{2}$$

Built upon the optimization of Equation 2, we further identify two additional issues that warrant consideration. 1). The synergy of *integrating prior information* and domain-invariant feature learning plays a crucial role. pre-trained (oracle) models can be used as priors (Junbum et al., 2022; Li et al., 2022) to regulate feature learning. 2). Issues regarding *invalid causality predicament* within the $\mathbf{Y}$-mapping come to light. When $\phi(\mathbf{X})$ is presumed to cause $\psi(\mathbf{Y})$ (i.e., $\phi(\mathbf{X}) \to \psi(\mathbf{Y})$ by maximizing $P(\phi(\mathbf{X}), \psi(\mathbf{Y}) \mid \mathcal{D})$), we should also suppress invalid causality $\psi(\mathbf{Y}) \to \phi(\mathbf{X})$ during invariant-feature learning (Refer to Equation 7 for derivation). Considering these findings, we propose a general objective for mDG, effectively relaxing the static target distribution assumption. It consists of four parts: **Aim1**- Learning domain-invariant representations and **Aim2**- Maximizing the posterior; with two regularization **Reg1**- Integrating prior information and the **Reg2**- Suppression of invalid causality. As a notable contribution, we redesign the conventional mDG paradigm and uniformly simplify the empirical objectives of various typical methods, as summarized in Table 1. We reveal that most previous works only partially optimize our proposed objective. Notations are shown in Table 2, and mathematical derivation details are provided in Appendix 4.

MDA (Hu et al., 2020) pioneered the relaxation of the static target distribution assumption, allowing that both $P(\mathbf{Y}|\mathcal{D})$ and $P(\mathbf{X}|\mathbf{Y}, \mathcal{D})$ change across the domain, yet without explicitly introducing a $\mathbf{Y}$-mapping function. We find that this objective can also be further termed as suppressing invalid causality regularization, which is similar to ours. RobustNet (Choi et al., 2021) uses instance-level mask elements in feature covariances and employs instance selective whitening loss, delivering efficacy comparable to suppressing invalid causality, but it may still be affected by varying $\mathbf{Y}$. CORAL (Sun & Saenko, 2016) exclusively minimizes shifts in feature covariance, implying a unique regularization on **Reg2**. SOTA methods such as MIRO (Junbum et al., 2022) and SIM-PLE (Li et al., 2022) propose learning similar features by "oracle" models as a substitute for learning domain-invariant representations for mDG. Worth mentioning, we counter MIRO's argument by confirming the persisting necessity of domain-invariant features, even under prior distribution, by theoretically deviating from minimizing the Generalized Jensen-Shannon Divergence (GJSD). Importantly, mere aggregation of the aforementioned objectives fails to yield a comprehensive general objective for mDG, as identified by our theoretical examination and empirical studies. For instance,

term $-H(P(\phi(\mathbf{X}|\mathcal{D})))$, coupled with prior knowledge utilization, could inadvertently precipitate performance degradation.

In summary, the general objective proposed for mDG tasks essentially comprises a weighted combination of **Four** terms referenced in Table 1, each term designated by an alias:

$$\min_{\phi,\psi} \underbrace{v_{A1} H(P(\phi(\mathbf{X}), \psi(\mathbf{Y}) \mid \mathcal{D}))}_{\text{GAim1}} + \underbrace{v_{A2}[H(P(\psi(\mathbf{Y}), \phi(\mathbf{X}))) + H(P(\mathbf{Y}, \psi(\mathbf{Y})))]}_{\text{GAim2}} \quad (3)$$
$$+ \underbrace{v_{R1} D_{\text{KL}}(P(\phi(\mathbf{X}), \psi(\mathbf{Y})) \| \mathcal{O})}_{\text{GReg1}} \underbrace{- v_{R2} H(P(\phi(\mathbf{X}) \mid \psi(\mathbf{Y}))) + H(P(\phi(\mathbf{X})))}_{\text{GReg2}}.$$

It allows us to relax the stable assumption of $\mathbf{Y}$ and achieves generalization across diverse mDG tasks. Empirically, we justify that **GAim1** and **GReg1** can be effectively revised by minimizing the Generalized Jensen-Shannon Divergence (GJSD) with prior knowledge between visible domains for optimization. Meanwhile, we derive an upper bound termed as an alignment Upper Bound with Prior of mDG (PUB). Regarding **GReg2**, which mitigates invalid causality scenarios, its general objective can be simplified by minimizing the Conditional Feature Shift (CFS), *i.e.,* the shift between unconditional and conditional features, which can be calculated by $\psi$. Our implementation can be readily integrated with existing mDG frameworks. Importantly, an array of experimental results echo the capability of the proposed general objective in augmenting performance across regression, segmentation, and classification tasks. From the ablation analysis, we find that inefficient optimization or unintended terms leveraged by previous methods, tested with uniform settings, result in degraded performance. Notably, our results that only used one pre-trained model as prior in classification even exceed the SOTA SIMPLE++, which employs 283 pre-trained models as an ensemble oracle.

## 2 RELATED WORK

**Domain generalization.** In order to learn better $D$-independent representations for domain generalization, DANN (Ganin et al., 2016) minimizes feature divergences between the source domains. CDANN Li et al. (2018d), CIDG (Li et al., 2018c), and MDA (Hu et al., 2020) additionally take conditions into consideration and aim to learn conditionally invariant features across domains. (Bui et al., 2021; Chattopadhyay et al., 2020; Junbum et al., 2022; Li et al., 2022) point out that learning invariant representation to source domains is insufficient for mDG. Therefore, MIRO (Junbum et al., 2022) and SIMPLE (Junbum et al., 2022) adopt pre-trained models as an oracle for seeking better general representations across various domains, including unseen target domains. In this paper, we show that these methods partially optimise our proposed objective, leading to sub-optimal results.

**DG assumptions.** In the literature, different assumptions are proposed to simplify the task as described by the original objective in Equation 1. One assumption is that the $P(\mathbf{Y}|\mathbf{X}, \mathcal{D})$ is stable while only marginal $P(\mathbf{X}|\mathcal{D})$ changes across domains (Shimodaira, 2000; Zhang et al., 2015). Zhang et al. (2013) point out that $\mathbf{X}$ is usually caused by $\mathbf{Y}$ thus $P(\mathbf{Y}|\mathcal{D})$ changes while $P(\mathbf{X}|\mathbf{Y}, \mathcal{D})$ is sable or $P(\mathbf{X}|\mathbf{Y}, \mathcal{D})$ changes but $P(\mathbf{Y}|\mathcal{D})$ stays stable, or a combination of both. Thus, MDA (Hu et al., 2020) allows that both $P(\mathbf{Y}|\mathbf{X}, \mathcal{D})$ and $P(\mathbf{X}|\mathcal{D})$ change across domains but needs selecting samples of each class for the calculation. Moreover, it considers no prior. This paper further relaxes these assumptions by extracting domain-invariant features in $\mathbf{X}, \mathbf{Y}$ without assuming that $\mathbf{Y}$ is invariant across domains.

**Using pre-trained models as the oracle model.** Previous methods such as MIRO (Junbum et al., 2022) have employed pre-trained models as the oracle to regularize $\phi$. SIMPLE (Li et al., 2022) employs at most 283 pre-trained models as an ensemble and adaptively composes the most suitable oracle model. This paper shows that their objectives are only partial of our proposed objectives, which lead to limited results.

## 3 A GENERAL MULTI-DOMAIN GENERALIZATION OBJECTIVE

This section presents the empirical losses used to implement Equation 3. Detailed derivation can be referred to in Appendix A.

**Learning of $\mathcal{D}$-independent conditional features under prior** is based on the generalized Jensen-Shannon divergence (GJSD) (Lin, 1991).

**Definition 1** (GJSD). *Given $J$ distributions, $\{P(\mathbf{Z}_j)\}_{j=1}^J$ and a corresponding probability weight vector $w$, $GJSD_w(\{P(\mathbf{Z}_j)\}_{j=1}^J)$ is defined as:*

$$\sum_{j=1}^J w_j D_{\mathrm{KL}}(P(\mathbf{Z}_j)\| \sum_{j=1}^J w_j P(\mathbf{Z}_j)) \equiv H(\sum_{j=1}^J w_j P(\mathbf{Z}_j)) - \sum_{j=1}^J w_j H(P(\mathbf{Z}_j)).$$

Our method addresses the standard scenario in which the weights are evenly distributed across domains: $w_1 = ... = w_N = 1/N$. To achieve $\phi(\mathbf{X}), \psi(\mathbf{Y}) \perp\!\!\!\perp \mathcal{D}$, minimizing domain gap between $P(\phi(\mathbf{X}_n), \psi(\mathbf{Y}_n))$ can be converted to minimizing GJSD across all domains:

$$\min_{\phi,\psi} GJSD(\{P(\phi(\mathbf{X}_n), \psi(\mathbf{Y}_n))\}_{n=1}^N) \tag{4}$$
$$\equiv \min_{\phi,\psi} H(P(\phi(\mathbf{X}), \psi(\mathbf{Y}) \mid \mathcal{D})) - \mathbb{E}[H(P(\phi(\mathbf{X}_n), \psi(\mathbf{Y}_n)))].$$

We further involve a prior knowledge distribution $\mathcal{O}$ under the consideration of a variational density model class $\mathcal{Q}$. Drawing upon (Cho et al., 2022), we have a variational upper bound:

$$GJSD(\{P(\phi(\mathbf{X}_n), \psi(\mathbf{Y}_n))\}_{n=1}^N) \leq H_c(\mathbb{E}[P(\phi(\mathbf{X}), \psi(\mathbf{Y})] \mid \mathcal{D}), \mathcal{O}) - a, \tag{5}$$

where $a \triangleq \sum_{n=1}^N H(P(\phi(\mathbf{X}_n), \psi(\mathbf{Y}_n)))$ is constant *w.r.t* $\phi, \psi$, hence ignored during optimization.

The novel GJSD variational upper bound, tied to domain generalization alignment (PUB) and derived from Equation 5, is:

$$\min_{\phi,\psi} PUB(\{P(\phi(\mathbf{X}_n), \psi(\mathbf{Y}_n))\}_{n=1}^N) \tag{6}$$
$$\triangleq \min_{\phi,\psi} H(\mathbb{E}[P(\phi(\mathbf{X}_n), \psi(\mathbf{Y}_n))]) + D_{\mathrm{KL}}(P(\phi(\mathbf{X}), \psi(\mathbf{Y}))\|\mathcal{O})) - a.$$

Minimizing PUB is the proposed objective for **GAim1** and **GReg1**. This implies that the methods like MIRO, solely minimizing GReg1, might result in substantial suboptimality, leaving the domain gap unresolved. We discuss two situations of $\mathcal{O}$ in Section 4.

**Suppressing invalid causality.** Figure 1 graphically demonstrates the causal diagram within our model. $\mathbf{Y}$ and $\psi(\mathbf{Y})$ are not assumed to cause $\phi(\mathbf{X})$, as $\mathbf{Y}$ is solely predicted from $\phi(\mathbf{X})$, despite their correlation. We aim to achieve learning domain-invariant representations by using $\mathbf{Y}$-mapping:

$$H(P(\phi(\mathbf{X}), \psi(\mathbf{Y}) \mid \mathcal{D}))$$
$$\leq H(P(\psi(\mathbf{Y}) \mid \mathcal{D})) + H(P(\phi(\mathbf{X}) \mid \psi(\mathbf{Y}), \mathcal{D}))$$
$$- H(P(\phi(\mathbf{X}) \mid \psi(\mathbf{Y}), \mathcal{D})) + H(P(\phi(\mathbf{X}) \mid \mathcal{D}))$$
$$= \mathbf{GAim1} + \mathbf{GReg2}. \tag{7}$$

Figure 1: Diagram of causality in the proposed method.

See more mathematical details in Appendix A. It unveils that **GAim1** tightens with employing **GReg2**, emphasizing the suppression of invalid causality, $\psi(\mathbf{Y}) \rightarrow \phi(\mathbf{X})$. Our ablation study also reveals that invalid causality may occur in invariant feature learning, and suppressing it leads to performance improvement.

We assume that $\phi(X), \psi(Y)$ in the RKSH follow Multivariate Gaussian-like Distributions which are denoted as $\mathcal{N}(\phi(\mathbf{X}); \mu_{\mathbf{X}}, \Sigma_{\mathbf{XX}}), \mathcal{N}(\psi(\mathbf{Y}); \mu_{\mathbf{Y}}, \Sigma_{\mathbf{YY}})$. Then, $P(\phi(\mathbf{X}) \mid \psi(\mathbf{Y}))$ follows $\mathcal{N}(\phi(\mathbf{X}) \mid \psi(\mathbf{Y}); \mu_{\mathbf{X}|\mathbf{Y}}, \Sigma_{\mathbf{XX}|\mathbf{Y}})$. **GReg2** can be simplified as:

$$H(\mathcal{N}(\phi(\mathbf{X}); \mu_{\mathbf{X}}, \Sigma_{\mathbf{XX}})) - H(\mathcal{N}(\phi(\mathbf{X}) \mid \psi(\mathbf{Y}); \mu_{\mathbf{X}|\mathbf{Y}}, \Sigma_{\mathbf{XX}|\mathbf{Y}})) = \frac{1}{2} \ln(\frac{|\Sigma_{\mathbf{XX}}|}{|\Sigma_{\mathbf{XX}|\mathbf{Y}}|}) \geq 0, \tag{8}$$

where the inequality stands owing to the *Condition Reducing Entropy*. This implies $H(\mathcal{N}(\phi(\mathbf{X}); \mu_{\mathbf{X}}, \Sigma_{\mathbf{XX}})) \geq H(\mathcal{N}(\phi(\mathbf{X}) \mid \psi(\mathbf{Y}); \mu_{\mathbf{X}|\mathbf{Y}}, \Sigma_{\mathbf{XX}|\mathbf{Y}}))$, deduced from $|\Sigma_{\mathbf{XX}}| \geq |\Sigma_{\mathbf{XX}|\mathbf{Y}}| \geq 0$, considering they are positive semi-definite.

Minimization of Equation 8 occurs iff $|\Sigma_{\mathbf{XX}}| = |\Sigma_{\mathbf{XX}|\mathbf{Y}}|$, reformulating the task as $min_{\phi,\psi}|\Sigma_{\mathbf{XX}}| - |\Sigma_{\mathbf{XX}|\mathbf{Y}}|$, where $\Sigma_{\mathbf{XX}|\mathbf{Y}} = \Sigma_{\mathbf{XX}} - \Sigma_{\mathbf{XY}}\Sigma_{\mathbf{YY}}^{-1}\Sigma_{\mathbf{YX}}$, per (Kay, 1993). Therefore, **GReg2** is simplified as minimizing Conditional Feature Shift (CFS):

$$\min_{\phi,\psi} |\Sigma_{\mathbf{XY}}\Sigma_{\mathbf{YY}}^{-1}\Sigma_{\mathbf{YX}}|. \tag{9}$$

**Empirical losses derivation.** We introduce the mapping $\psi$ to relax the static target distribution. The implementation of $\psi$ varies across tasks, utilizing MLPs for classification and regression, and ResNet-50 for segmentation. To promote a consistent latent space, the mapped $\psi(\mathbf{Y})$ retains the same dimension as that of $\phi(\mathbf{X})$. $\psi(\mathbf{Y})$ and $\phi(\mathbf{X})$ are separately fed into $\mathcal{C}$ for making predictions and obtaining $\mathcal{L}_{A2}$ for posterior maximization:

$$\mathcal{L}_{A2}(\mathcal{C}, \phi, \psi) = H_c(\phi(\mathbf{X}), \mathbf{Y}) + H_c(\psi(\mathbf{Y}), \mathbf{Y}). \tag{10}$$

To mitigate domain shifts and learn domain invariance, we minimize cross-domain conditional feature distribution discrepancies. Specifically, the mean and variance of the joint distribution of $(\phi(\mathbf{X}), \psi(\mathbf{Y}))$ in each domain are estimated using VAE encoders. Consider $n$-pairs means and variance of $n$ domains, we derive a joint Gaussian distribution expression $P(\phi(\mathbf{X}_n), \psi(\mathbf{Y}_n)) \triangleq \mathcal{N}(\boldsymbol{x}_n, \boldsymbol{y}_n; \mu_n, \Sigma_n)$. Accordingly, we establish $\mathbb{E}[P(\phi(\mathbf{X}_n), \psi(\mathbf{Y}_n))] \triangleq \mathcal{N}(\bar{\boldsymbol{x}}, \bar{\boldsymbol{y}}; \bar{\mu}, \bar{\Sigma})$ where $\bar{\mu} = \mathbb{E}[\mu_n], \bar{\Sigma} = \mathbb{E}[\Sigma_n]$. Base on PUB in Equation 6, we introduce $\mathcal{L}_{A1}$ to minimize the conditional feature gap across domains:

$$\mathcal{L}_{A1}(\phi) = \sum_{i=1}^{n} (\log |\Sigma_i| + ||\bar{\mu} - \mu_i||^2_{\Sigma_i^{-1}}). \tag{11}$$

To integrate prior information, similar to MIRO, we utilize VAE encoders to capture the means and variances of $\mathbf{X}$: $P(\phi(\mathbf{X})) \triangleq \mathcal{N}(\boldsymbol{x}; \mu_x, \Sigma_x)$ and the output features $x_{\mathcal{O}}$ form $\mathcal{O}$. Given that $\mathcal{O}$ preserves the correlation between $\mathbf{X}$ ($\phi(\mathbf{X})$) and $\mathbf{Y}$ ($\psi(\mathbf{Y})$), and is frozen during training, $\mathbf{Y}, \psi(\mathbf{Y})$ is omitted in empirical loss. We propose $\mathcal{L}_{R1}$ to minimize the divergence between features and $\mathcal{O}$:

$$\mathcal{L}_{R1}(\phi) = \log |\Sigma_x| + ||x_{\mathcal{O}} - \mu_x||^2_{\Sigma_x^{-1}}. \tag{12}$$

For suppressing the invalid causality, derived from Equation 9, the loss is designed to minimize the CFS, and is thus defined as:

$$\mathcal{L}_{R2}(\phi) = ||\Sigma_{\mathbf{XY}} \Sigma_{\mathbf{YY}}^{-1} \Sigma_{\mathbf{YX}}||_2, \tag{13}$$

where $\Sigma_{\mathbf{XY}} = \mathbb{E}[\phi(\mathbf{X}) - \mathbb{E}[\phi(\mathbf{X})]]^\top (\phi(\mathbf{Y}) - \mathbb{E}[\phi(\mathbf{Y})]]$, and a similar calculation process is done for $\Sigma_{\mathbf{YY}}$ and $\Sigma_{\mathbf{YX}}$. The final loss is a weighted combination of the above losses:

$$\mathcal{L}(\mathcal{C}, \phi, \psi) = v_{A1}\mathcal{L}_{A1} + v_{A2}\mathcal{L}_{A2} + v_{R1}\mathcal{L}_{R1} + v_{R2}\mathcal{L}_{R2}. \tag{14}$$

Detailed hyper-parameters settings can be seen in Appendix C.

## 4 VALIDATING PROPOSED OBJECTIVE: THEORETICAL ANALYSIS

In this section, we theoretically validate our objective function's efficacy. By showing the connections between our proposed objectives and previous ones, we reveal that many previous mDG endeavors partially optimize the proposed objective. Detailed understanding of previous objectives can be referred to in Table 1 and Appendix B.

**Using $\psi$ v.s. not using $\psi$.** Previous works rarely employed $\psi$ to map $\mathbf{Y}$, whereas we show its benefits for mDG tasks. Employing *Jensen's inequality*, we obtain $H(\mathbb{E}[P(\phi(\mathbf{X}_n), \psi(\mathbf{Y}_n))]) \geq H(\mathbb{E}[P(\phi(\mathbf{X}_n), \mathbf{Y}_n)])$. When other objectives remain the same, we compare the model with parameters $\theta^\psi$ optimized via the $\psi$ mapping, against another model without $\psi$ using parameters $\theta^{n\psi}$:

$$\sup R(\theta^{n\psi}) \geq \sup R(\theta^\psi). \tag{15}$$

The equivalence is valid only if $\psi$ serves as a bijection, a condition prevalent in practical scenarios like classification. Thus, this mapping does not hinder model performance in classification tasks. It also implies that using $\psi(\mathbf{Y})$ can lower generalization risks after optimization, especially when $\mathbf{Y}$ contains features dependent on $\mathcal{D}$. This could potentially yield superior generalization in segmentation and regression tasks. Detailed proof can be seen in Appendix A.

**Remark 1** (Importance of $\mathbf{Y}$ mapping $\psi$)**.** *Besides relaxing the static distribution assumption of $\mathbf{Y}$, $\psi$ conveys two other notable benefits: 1). $\mathbf{X}$ and $\mathbf{Y}$ may originate from different sample spaces with distinct shapes. By applying mappings, $\psi(\mathbf{Y})$ can be adapted to the same shape as $\phi(\mathbf{X})$. In practice, concatenating $\phi(\mathbf{X})$ and $\psi(\mathbf{Y})$ is often used as input for VAE encoders to capture $P(\psi(\mathbf{Y}), \phi(\mathbf{X}))$. 2). The derivation of Equation 9 requires the computation of covariance, which mandates that two variables occupy the same sample space - a condition fulfilled by applying $\psi(\mathbf{Y})$.*

Figure 2: Diagram of constructing the toy dataset.

**Incorporating conditions leads to lower generalization risk on learning invariant representations.** A few past works (Ganin et al., 2016; Sun & Saenko, 2016) minimize domain gaps between features without condition consideration. Its objective for **Aim1** is:

$$H(P(\phi(\mathbf{X}_n))) \leq H(P(\phi(\mathbf{X}_n))) + H(P(\psi(\mathbf{Y}_n) \mid \phi(\mathbf{X}_n))) = \mathbf{GAim1}. \tag{16}$$

While the other objectives are identical, we consider a model with parameters $\theta^{nc}$, trained with $\min_{\psi} H(P(\phi(\mathbf{X}_n)))$, against another model with $\theta^c$ parameters, trained with **GAim1**. In this scenario, their empirical risks satisfy:

$$\sup R(\theta^{nc}) \geq \sup R(\theta^c). \tag{17}$$

See the mathematical details in Appendix A. This reveals that without condition consideration, the minimization of generalization risk is merely partial due to the overlooked risk correlated to $\mathbf{Y}$. Additional evidence supporting the importance of condition consideration is provided by CDANN (Li et al., 2018d) and CIDG (Li et al., 2018c). Our experiments, conducted through a uniform implementation, also lend support to it.

**Effect of oracle model $\mathcal{O}$.** As stated by MIRO (Junbum et al., 2022) and SIMPLE (Li et al., 2022), a generalized $\mathcal{O}$ comprising both seen and unseen domains yields significant improvements. During the derivation of Equation 6, we find that the disregard **GAim1** term in MIRO (Junbum et al., 2022) and SIMPLE (Li et al., 2022) may result in inferior outcomes to our proposed objective.

**Remark 2** (Synergy of learning invariance, integrating prior knowledge and suppressing invalid causally). *For readability, we've divided the overall mDG objective into four aspects, despite all terms being interconnected. Specifically, as shown by PUB in Equation 6, **GReg1** collaborating with **GAim1** brings more performance gains than the case when it is solely applied. Moreover, Equation 7 shows that **GAim1** is made tighter by combining with **GReg2**, underscoring the significance of combining learning invariance, integrating prior knowledge, and suppressing invalid causally. It also suggests that all terms are synergistic and contribute together to improved results.*

Validating our assertions via experiments, Section 4.4 ablation study finds that simple cross-domain covariance limitation (**GReg2**) cannot ensure improved results with prior knowledge.

## 4.1 TOY EXPERIMENTS: REGRESSION

We perform a regression task on synthetic data to illustrate the impact of using $\psi$, showcasing its potential for superior results if $\psi$ is not bijective.

**Synthetic data.** Figure 2 illustrates the construction of synthetic data, built on $\mathbf{X}$-$\mathbf{Y}$ pair latent features with a linear relationship, ensuring invariant existence. To better explore this issue, we created four distinct data groups: without and with distribution shift, used affine or squared and cubed transformations as domain-conditioned transformations, and their cross combinations. More description can be seen in Appendix C.

Table 3: Toy experiments: MSE losses on testing set. $\mathcal{L}_{PUB}$ and $\psi$ are cumulatively added to the basic ERM. Best results are highlighted as **bold**. DCDS denotes domain-conditioned distribution shift.

| | Affine transformations | | |
| --- | --- | --- | --- |
| | ERM | $+\mathcal{L}_{A1}(\phi)$ | $+\mathcal{L}_{A1}(\phi, \psi)$ |
| No DCDS | 0.3485 | 0.3537 | **0.3369** |
| With DCDS | 0.4144 | 0.2290 | **0.1777** |
| | Squared and cubed transformations | | |
| | ERM | $+\mathcal{L}_{A1}(\phi)$ | $+\mathcal{L}_{A1}(\phi, \psi)$ |
| No DCDS | 1.5150 | 0.4652 | **0.3370** |
| With DCDS | 0.8720 | 1.5868 | **0.8241** |

**Experimental setup.** We use two of three constructed domains for training and validation and the last one for testing. Validation and test losses are calculated by MSE. To maintain fairness, all experiments adopt the same network which is selected by the best validation results. Learning aims to find invariant hidden features of $X, Y$ while preserving predictive ability from unseen $X$ to $Y$.

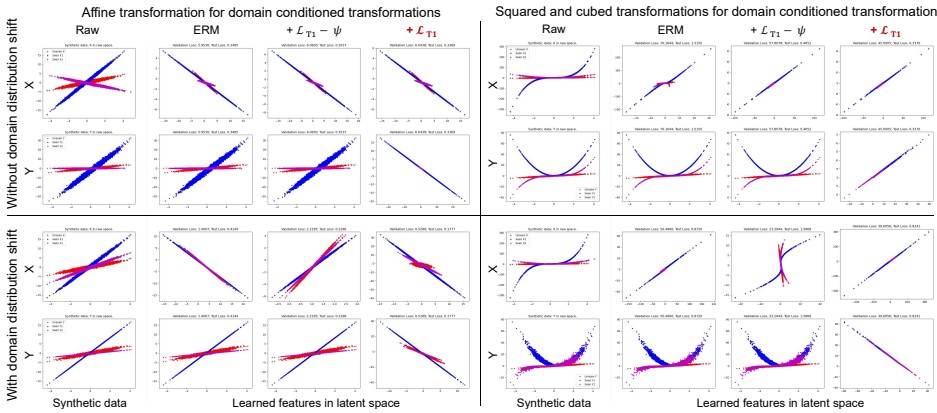

Figure 3: Toy experiments: Visualization of learned latent representations of different methods.

**Results.** Toy experiment results are reported in Table 3, which are also visualized in Figure 3. It is observed that across all settings, employing $\psi$ with $\mathcal{L}_{A1}$ yields superior results, outperforming ERM and ERM+$\mathcal{L}_{A1}(\phi)$ without $\psi$, validating the enhanced generalization effect brought by utilizing $\psi$ whenever $\mathbf{Y}$ varies per domain, supporting Equation 15. Figure 3 shows that $\psi$ does learn the invariant representations for $Y$ to relax previous $Y$-invariant assumption. Specifically, learning the invariance of $Y$ with $\psi$ results in superior invariant representations as the latent representations of $X, Y$ are primarily linear, aligning with $X$ and $Y$'s linear relationship during data construction. The bottom-left figures reveal that though ERM has learned the most invariant $\phi(X)$, it suffers the worst test loss, indicating that a well-learned invariant $\phi(X)$ is not sufficient when $Y$ also has domain-dependent traits. The results also suggest that assuming that $Y$ vary across domains, using $\mathcal{L}_{A1}$ without $\psi$ may not yield superior results.

## 4.2 SEGMENTATION ON REALISTIC AND SYNTHETIC DATASETS

**Experimental setup.** We follow the experimental setup of RobustNet (Choi et al., 2021) for mDG segmentation experiments, particularly using DeepLabV3+ (Chen et al., 2018) as the semantic segmentation model architecture, with ResNet-50 backbone and SGD optimizer, initial learning rate of 1e-2, and momentum of 0.9. As shown in Table 1, RobustNet's objective is equivalent to using **GAim2** and **GReg2**. Consistent with previous methods, mIoU serves as our evaluation metric. Datasets comprise real-world datasets (Cityscapes (Cordts et al., 2016), BDD-100K (Yu et al., 2020), Mapillary (Neuhold et al., 2017)) and synthetic datasets (GTAV (Richter et al., 2016), SYNTHIA (Ros et al., 2016)). Specifically, we train a model on GTAV and Cityscapes, testing on other datasets. Table 4 shows formulations of each term.

Table 4: Notations for terms

| | |
|---|---|
| **GAim2** | $H(P(\psi(\mathbf{Y}) \mid \phi(\mathbf{X}))) + H(P(\mathbf{Y} \mid \psi(\mathbf{Y})))$ |
| **GReg1** | $D_{\mathrm{KL}}(\mathbb{E}[P(\phi(\mathbf{X}_n), \mathbf{Y}_n)]\|\mathcal{O}))$ |
| **iAim1** | $H(\mathbb{E}[P(\phi(\mathbf{X}_n))])$ |
| **GAim1** | $H(\mathbb{E}[P(\phi(\mathbf{X}_n), \mathbf{Y}_n)])$ |
| **iReg2** | $-H(\mathbb{E}[P(\phi(\mathbf{X}_n))]) + H(P(\phi(\mathbf{X})))$ |
| **GReg2** | $-H(P(\phi(\mathbf{X}) \mid \psi(\mathbf{Y}))) + H(P(\phi(\mathbf{X})))$ |

Table 5: Segmentation experiments: Comparison of mIoU(%). The models are trained on multiple synthetic domains. We use the initials to denote each dataset.

| Models (G + S) | C | B | M | Avg. |
|---|---|---|---|---|
| Baseline | 35.46 | 25.09 | 31.94 | 30.83 |
| IBN-Net | 35.55 | 32.18 | 38.09 | 35.27 |
| RobustNet (**GAim2**, **GReg2**) | 37.69 | 34.09 | 38.49 | 36.76 |
| **+GAim1** | 38.58 | 34.72 | 39.11 | 37.47 |
| **+GReg1** | 38.13 | **35.02** | 39.29 | 37.48 |
| **+GAim1+GReg1** | **38.62** | 34.71 | **39.63** | **37.65** |

**Results.** Table 5 shows the efficacy of our proposed objective in segmentation tasks upon introducing $\psi$. Ablation results highlight that using $\psi$ alongside **GAim1** can enhance baseline performance, experimentally substantiating that the introduction of $\psi$, in relaxing assumptions, boosts performance for better generalization. Using **GReg1** alone also improves average mIoU. Importantly, the most enhancement in average mIoU is observed when **GReg1** and **GAim1** are used together, which finds validation in the PUB derivation in Equation 6.

Table 6: Classification results: Our method compared with previous non-ensemble and ensemble mDG methods. The best results for each group are highlighted as **bold**.

| | PACS | VLCS | OfficeHome | TerraInc | DomainNet | Avg. |
|---|---|---|---|---|---|---|
| **Non-ensemble methods** | | | | | | |
| CDANN (Li et al., 2018d) | 82.6±0.9 | 77.5±0.1 | 65.8±1.3 | 45.8±1.6 | 38.3±0.3 | 62.0 |
| DANN (Ganin et al., 2016) | 83.6±0.4 | 78.6±0.4 | 65.9±0.6 | 46.7±0.5 | 38.3±0.1 | 62.6 |
| ERM (Vapnik, 1998) | 84.2±0.1 | 77.3±0.1 | 67.6±0.2 | 47.8±0.6 | 44.0±0.1 | 64.2 |
| CORAL (Sun & Saenko, 2016) | 86.2±0.3 | 78.8±0.6 | 68.7±0.3 | 47.6±1.0 | 41.5±0.1 | 64.5 |
| Use ResNet-50 (He et al., 2016) as oracle model. | | | | | | |
| MIRO (Junbum et al., 2022) | 85.4±0.4 | 79.0±0.3 | 70.5±0.4 | 50.4±1.1 | 44.3±0.2 | 65.9 |
| **Ours** | 85.6±0.3 | **79.2**±0.3 | **70.7**±0.2 | **51.1**±0.9 | **44.6**±0.1 | **66.3** |
| Use RegNetY-16GF (Singh et al., 2022) as oracle model. | | | | | | |
| MIRO | **97.4**±0.2 | 79.9±0.6 | 80.4±0.2 | 58.9±1.3 | 53.8±0.1 | 74.1 |
| **Ours** | 97.3±0.1 | **82.4**±0.6 | **80.8**±0.6 | **60.7**±1.8 | **54.6**±0.1 | **75.1** |
| **Ensemble methods** | | | | | | |
| | PACS | VLCS | OfficeHome | TerraInc | DomainNet | Avg. |
| Use multiple oracle models. | | | | | | |
| SIMPLE (Li et al., 2022) | 88.6±0.4 | 79.9±0.5 | 84.6±0.5 | 57.6±0.8 | 49.2±1.1 | 72.0 |
| SIMPLE++ (Li et al., 2022) | **99.0**±0.1 | **82.7**±0.4 | **87.7**±0.4 | **59.0**±0.6 | **61.9**±0.5 | **78.1** |
| Use ResNet-50 (He et al., 2016) as oracle model. | | | | | | |
| MIRO + SWAD | **88.4**±0.1 | 79.6±0.2 | 72.4±0.1 | 52.9±0.2 | 47.0±0.0 | 68.1 |
| **Ours + SWAD** | **88.4**±0.1 | **79.6**±0.1 | **72.5**±0.2 | **53.0**±0.7 | **47.3**±0.1 | **68.2** |
| Use RegNetY-16GF (Singh et al., 2022) as oracle model. | | | | | | |
| MIRO + SWAD | 96.8±0.2 | 81.7±0.1 | 83.3±0.1 | 64.3±0.3 | 60.7±0.0 | 77.3 |
| **Ours + SWAD** | **97.9**±0.3 | **82.2**±0.3 | **84.7**±0.2 | **65.0**±0.2 | **61.3**±0.2 | **78.2** |

## 4.3 CLASSIFICATION ON BATCHMARK DATASETS

**Experimental setup.** We operate on the DomainBed suite (Gulrajani & Lopez-Paz, 2020) and leverage standard leave-one-out cross-validation as an evaluation method. We experiment on 5 real-world benchmark datasets, including PACS (4 domains, 9,991 samples, 7 classes) (Li et al., 2017), VLCS (4 domains, 10,729 samples, 5 classes) (Fang et al., 2013), OfficeHome (4 domains, 15,588 samples, 65 classes) (Venkateswara et al., 2017), TerraIncognita (TerraInc, 4 domains, 24,778 samples, 10 classes) (Beery et al., 2018), and DomainNet (6 domains, 586,575 samples, 345 classes) (Peng et al., 2019). The results are the averages from three trials of each experiment. Following MIRO, two backbones are used for the training (ResNet-50 (He et al., 2016) pre-trained in the ImageNet (He et al., 2016) and RegNetY-16GF backbone with SWAG pre-training (Singh et al., 2022)). The backbones are trained with our proposed objective barely and further with SWAD (Cha et al., 2021), respectively. See Appendix C for more experimental details.

**Results.** Table 6 displays the results of non-ensemble algorithms and ensemble algorithms that employ pre-trained models as oracle models. Specifically, our proposed objectives demonstrate more substantial improvements when a higher-quality pre-trained oracle model ($\mathcal{O}$) is applied. When employing ResNet-50 model, our approach yields average improvements of approximately 0.3% and 0.1% without and with SWAD, respectively, compared to MIRO. In contrast, when RegNetY-16GF serves as an oracle, our proposed objectives result in significant average improvements of 1.1% and 0.9% without and with SWAD, respectively. Remarkably, our approach outperforms 0.1% more than the SOTA method, SIMPLE++, which relies on an ensemble of 283 pre-trained models as oracle models, whereas ours only engages a single pre-trained model. Overall, these results strongly support our objective's effectiveness in classification tasks. See additional results in Appendix D.

## 4.4 ABLATION STUDIES

**Experimental setup.** In the ablation studies, we test varied terms (see Table 7) combinations on the HomeOffice dataset using SWAG pre-training (Singh et al., 2022) and SWAD (Cha et al., 2021). Every experiment is repeated in three trials, sharing the same hyper-parameter settings for evaluation. See Appendix C for more experimental details.

**Results.** Table 7 presents ablation study results. The first column denotes previous methods equivalent to term combinations. The main findings are as follows.

1). Previous methods that partially utilize our proposed objectives often yield suboptimal results. By eliminating other factors, it can be seen that employing our proposed full objectives offers the most significant improvements, while previous objectives may lead to inferior results.

Table 7: Ablation studies: Results of using different combinations of terms on HomeOffice. Imp. denotes Improvement that gained form **GAim2** and **GAim2 + GReg1**, respectively.

| Used objectives | art | clipart | product | real | avg | Imp. |
|---|---|---|---|---|---|---|
| Without $\mathcal{O}$ (**GReg1**) | | | | | | |
| **GAim2** (ERM) | 78.4±0.7 | 68.3±0.5 | 85.8±0.4 | 85.8±0.3 | 79.6±0.2 | 0.0 |
| **GAim2 + iAim1** (DANN) | 79.1±1.0 | 68.6±0.0 | 85.6±0.8 | 86.1±0.5 | 79.8±0.2 | +0.2 |
| **GAim2 + GAim1** (CDANN, CIDG) | 79.1±0.7 | 69.1±0.1 | 85.7±0.5 | 86.3±0.6 | 79.9±0.4 | +0.3 |
| **GAim2 +iReg2** (CORAL+$\psi$) | 79.1±0.1 | 69.9±0.4 | 86.0±0.1 | 86.3±0.4 | 80.3±0.2 | +0.7 |
| **GAim2 + GReg2** | 79.2±0.1 | **69.9**±1.4 | 86.1±0.5 | 86.1±0.1 | 80.3±0.3 | +0.7 |
| **GAim2 + GAim1 + GReg2** (MDA+$\psi$) | **79.5**±1.1 | 69.2±1.2 | **86.2**±0.2 | **86.5**±0.2 | **80.3**±0.0 | **+0.7** |
| With $\mathcal{O}$ (**GReg1**) | | | | | | |
| **GAim2 + GReg1** (MIRO, SIMPLE) | 83.2±0.6 | 72.6±1.1 | 89.9±0.5 | 90.2±0.1 | 84.0±0.2 | 0.0 |
| **GAim2 + GReg1 +iAim1** | 83.4±0.5 | 73.1±0.8 | 89.7±0.4 | 90.1±0.3 | 84.1±0.2 | +0.1 |
| **GAim2 + GReg1 + GAim1** | 83.7±0.3 | 74.0±0.6 | 90.1±0.3 | 90.3±0.2 | 84.5±0.2 | +0.4 |
| **GAim2 + GReg1 + iReg2** | 82.9±0.5 | 72.5±0.3 | **90.3**±0.3 | 90.0±0.3 | 83.9±0.1 | -0.1 |
| **GAim2 + GReg1 + GReg2** | 83.4±0.2 | 72.3±0.2 | 90.1±0.3 | 90.1±0.3 | 84.0±0.2 | +0.0 |
| **GAim2 + GReg1 + GAim1 + GReg2** (Ours) | **84.1**±0.2 | **74.3**±0.9 | 89.9±0.4 | **90.6**±0.1 | **84.7**±0.2 | **+0.7** |

2). The effectiveness of using conditions. By conducting uniform implementation and testing, we can observe that the use of conditions yields superior results compared to the unconditional approach. This observation aligns with Equation 17, suggesting that aligning conditional features across domains leads to improved generalization. Note that **iAim1** is the unconditional version of **GAim1**. The disparity in performance between CDANN and DANN in Table 6 might be attributed to differences in their implementation details.

3). Learning invariance is crucial, regardless of whether integrating prior knowledge. Evidently, learning invariance facilitates improvement whether prior is applied or not, as validated in the PUB derivation in Equation 6. This contradicts MIRO's argument that achieving similar representations to a prior can replace the need for learning invariance.

4). Impacts of using prior. The significant improvement owes to the use of a pre-trained oracle model ($\mathcal{O}$) preserving correlations between $\mathbf{X}$ and $\mathbf{Y}$ - a concept validated by MIRO and SIMPLE. However, utilizing our full set of objectives can further enhance this improvement by an additional 0.7%. Notably, the invalid causality may not work when using prior knowledge, while the invariance across domains is not permitted. We hypothesize that such invalid causality is inherently eliminated within a 'good' feature space obtained by $\mathcal{O}$, but may be reintroduced when we minimize the domain gap with $\mathcal{O}$. Thus, using the full objective can synergistically produce optimal results.

**Constraining only the covariance shifts of features across domains (GReg2) does not guarantee better results when prior knowledge is available.** We find that using the objectives of CORAL performs better than DANN, CDANN, and CIDG. The results suggest that considering the covariance shifts of features does lead to improvements, which we hypothesize are primarily driven by $H(P(\phi(\mathbf{X})))$. However, when a large pre-trained oracle model ($\mathcal{O}$) is provided, the performance actually degrades. This implies that the use of $\mathcal{O}$ implicitly minimizes the covariance shifts of features across domains. Under this scenario, the unexpected effect of $-H(P(\phi(\mathbf{X}|\mathcal{D})))$ hinders improvement, while the benefits brought by $H(P(\phi(\mathbf{X})))$ are diminished by the use of prior knowledge. In contrast, **GReg2** continues to yield improvements. This suggests that our objective is more versatile and suitable for various situations

## 5 CONCLUSION

In this paper, by relaxing the static distribution assumption of $\mathbf{Y}$ through a learnable mapping $\psi$, we propose a general objective that consists of minimizing conditional features domain gaps, incorporating prior knowledge, maximizing a posterior, and suppressing invalid causality. Our proposed objective is applicable to diverse mDG tasks including regression, segmentation, and classification. Empirically, we design a suite of losses to achieve the overall general objective, adaptable across various frameworks. Extensive experiments validate the viability of our objective across applications where previous objectives may yield suboptimal results. compared to ours. Both theoretical analyses and empirical results demonstrate the synergistic effect of distinct terms in the proposed objective. Simplistically, we assume equal domain weights whilst minimizing GJSD, presenting the future scope for dealing with imbalance situations triggering unequal domain weights.

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

## A MORE MATHEMATICAL DETAILS OF OUR METHOD

### A.1 MORE DETAILS ABOUT TABLE 1

To better understand, we simplify some notations in Table 8. We present the simplified notations and their corresponding origins in Table 8.

Table 8: Supplemental notations for Table 1. Refined notations and their original formulations are reported. The original formulations are highlighted as blue.

| | Learning domain invariant representations | |
| | Aim1: Learning domain invariance | Reg1: Integrating prior |
|---|---|---|
| DANN | $\min_\phi H(P(\phi(\mathbf{X}) \mid \mathcal{D}))$ 
 $\min_\phi H(\mathbb{E}[P(\phi(\mathbf{X}_n))])$ | None |
| CDANN, CIDG, MDA | $\min_\phi H(P(\phi(\mathbf{X}), \mathbf{Y} \mid \mathcal{D}))$ 
 $\min_\phi H(\mathbb{E}[P(\phi(\mathbf{X}_n), \psi(\mathbf{Y}_n))])$ | None |
| **Ours** | $\min_{\phi,\psi} H(P(\phi(\mathbf{X}), \psi(\mathbf{Y}) \mid \mathcal{D}))$ 
 $\min_{\phi,\psi} H(\mathbb{E}[P(\phi(\mathbf{X}_N), \psi(\mathbf{Y}_n)])$ | $\min_{\phi,\psi} D_{\mathrm{KL}}(P(\phi(\mathbf{X}), \psi(\mathbf{Y}))\|\mathcal{O})$ |

| | Maximizing A Posterior between representations and targets | |
| | Aim2: Maximizing A Posterior (MAP) | Reg2: Suppressing invalid causality |
|---|---|---|
| CORAL | $\min_\phi H(P(\mathbf{Y} \mid \phi(\mathbf{X})))$ | $\min_\phi -H(P(\phi(\mathbf{X} \mid \mathcal{D}))) + H(P(\phi(\mathbf{X})))$ 
 $\min_\phi -H(\mathbb{E}[P(\phi(\mathbf{X}_n))]) + H(P(\phi(\mathbf{X})))$ |

### A.2 DERIVATION DETAILS OF PUB

**Details for Equation 4.**

We denote $P_{mix} \triangleq \sum_n w_n P(\phi(\mathbf{X}_n), \psi(\mathbf{Y}_n))$. Therefore for GJSD, we have:

$$
\begin{aligned}
& GJSD(\{P(\phi(\mathbf{X}_n), \psi(\mathbf{Y}_n))\}_{n=1}^N) \qquad\qquad (18) \\
&= \sum_n w_n KL(P(\phi(\mathbf{X}_n), \psi(\mathbf{Y}_n))\|P_{mix}) \\
&= \sum_n w_n[H_c(P(\phi(\mathbf{X}_n), \psi(\mathbf{Y}_n)), P_{mix}) - H(P(\phi(\mathbf{X}_n), \psi(\mathbf{Y}_n)))] \\
&= \sum_n w_n H_c(P(\phi(\mathbf{X}_n), \psi(\mathbf{Y}_n)), P_{mix}) - \sum_n w_n H(P(\phi(\mathbf{X}_n), \psi(\mathbf{Y}_n))) \\
&= \sum_n w_n \int_{\phi(\mathbf{X}_n), \psi(\mathbf{Y}_n)} -P(x,y) \ln P_{mix}(x,y) d(x,y) - \sum_n w_n H(P(\phi(\mathbf{X}_n), \psi(\mathbf{Y}_n))) \\
&= \int_{\phi(\mathbf{X}_n), \psi(\mathbf{Y}_n)} -\sum_n w_n P(x,y) \ln P_{mix}(x,y) d(x,y) - \sum_n w_n H(P(\phi(\mathbf{X}_n), \psi(\mathbf{Y}_n))) \\
&= \int_{\phi(\mathbf{X}_n), \psi(\mathbf{Y}_n)} -P_{mix}(x,y) \ln P_{mix}(x,y) d(x,y) - \sum_n w_n H(P(\phi(\mathbf{X}_n), \psi(\mathbf{Y}_n))) \\
&= H(P_{mix}) - \sum_n w_n H(P(\phi(\mathbf{X}_n), \psi(\mathbf{Y}_n))).
\end{aligned}
$$

Therefore, the minimization of GJSD can be written as follows:

$$
\begin{aligned}
& \min_{\phi,\psi} GJSD(\{P(\phi(\mathbf{X}_n), \psi(\mathbf{Y}_n))\}_{n=1}^N) \\
&\equiv \min_{\phi,\psi} H(\mathbb{E}[P(\phi(\mathbf{X}_n), \psi(\mathbf{Y}_n))]) - \mathbb{E}[H(P(\phi(\mathbf{X}_n), \psi(\mathbf{Y}_n)))], \\
&\equiv \min_{\phi,\psi} H(P(\phi(\mathbf{X}), \psi(\mathbf{Y}) \mid \mathcal{D})) - \mathbb{E}[H(P(\phi(\mathbf{X}_n), \psi(\mathbf{Y}_n)))].
\end{aligned}
$$

**Details for Equation 5.** Taking into account $\mathcal{O}$, similar to (Cho et al., 2022), we have the upper bound for GJSD as:

$$
\begin{aligned}
&GJSD(\{P(\phi(\mathbf{X}_n), \psi(\mathbf{Y}_n))\}_{n=1}^N) \qquad\qquad\qquad (19)\\
&=H_c(P_{mix}, \mathcal{O}) - H_c(P_{mix}, \mathcal{O}) + H(P_{mix}) - \sum_n w_n H(P(\phi(\mathbf{X}_n), \psi(\mathbf{Y}_n)))\\
&=H_c(P_{mix}, \mathcal{O}) - D_{\mathrm{KL}}(P_{mix}\|\mathcal{O}) - \sum_n w_n H(P(\phi(\mathbf{X}_n), \psi(\mathbf{Y}_n)))\\
&\leq H_c(P_{mix}, \mathcal{O}) - \sum_n w_n H(P(\phi(\mathbf{X}_n), \psi(\mathbf{Y}_n))).
\end{aligned}
$$

For the standard situation where $w_1 = w_2 = ... = w_n = 1/N$, we further have:

$$
\begin{aligned}
&GJSD(\{P(\phi(\mathbf{X}_n), \psi(\mathbf{Y}_n))\}_{n=1}^N)\\
&\leq H_c(P_{mix}, \mathcal{O}) - \sum_n w_n H(P(\phi(\mathbf{X}_n), \psi(\mathbf{Y}_n)))\\
&=H_c(\mathbb{E}[P(\phi(\mathbf{X}_n), \psi(\mathbf{Y}_n))], \mathcal{O}) - \mathbb{E}[H(P(\phi(\mathbf{X}_n), \psi(\mathbf{Y}_n)))]. \qquad (20)
\end{aligned}
$$

**Details for Equation 6.** The above bound can be further reformed as:

$$
\begin{aligned}
&H_c(\mathbb{E}[P(\phi(\mathbf{X}_n), \psi(\mathbf{Y}_n))], \mathcal{O}) - a\\
&=H(\mathbb{E}[P(\phi(\mathbf{X}_n), \psi(\mathbf{Y}_n))]) + D_{\mathrm{KL}}(\mathbb{E}[P(\phi(\mathbf{X}_n), \psi(\mathbf{Y}_n))]\|\mathcal{O}) - a,\\
&=H(\mathbb{E}[P(\phi(\mathbf{X}_n), \psi(\mathbf{Y}_n))]) + D_{\mathrm{KL}}(P(\phi(\mathbf{X}), \psi(\mathbf{Y}))\|\mathcal{O}) - a. \qquad (21)
\end{aligned}
$$

**Derivation details of Equation 7.**

$$
\begin{aligned}
&H(P(\phi(\mathbf{X}), \psi(\mathbf{Y}) \mid \mathcal{D})) = H(P(\psi(\mathbf{Y}) \mid \mathcal{D})) + H(P(\phi(\mathbf{X}) \mid \psi(\mathbf{Y}), \mathcal{D}))\\
&\leq H(P(\psi(\mathbf{Y}) \mid \mathcal{D})) + H(P(\phi(\mathbf{X}) \mid \mathcal{D}))\\
&= H(P(\psi(\mathbf{Y}) \mid \mathcal{D})) \underbrace{+H(P(\phi(\mathbf{X}) \mid \psi(\mathbf{Y}), \mathcal{D})) - H(P(\phi(\mathbf{X}) \mid \psi(\mathbf{Y}), \mathcal{D}))}_{=0} +H(P(\phi(\mathbf{X}) \mid \mathcal{D}))\\
&= \mathbf{GAim1} - H(P(\phi(\mathbf{X}) \mid \psi(\mathbf{Y}), \mathcal{D})) + H(P(\phi(\mathbf{X}) \mid \mathcal{D})).
\end{aligned}
$$

It can be reformed as:

$$
\begin{aligned}
&\mathbf{GAim1} - H(P(\phi(\mathbf{X}) \mid \psi(\mathbf{Y}), \mathcal{D})) + H(P(\psi(\mathbf{Y}), \mathcal{D})) + H(P(\phi(\mathbf{X}) \mid \mathcal{D}))\\
&\quad \underbrace{-H(P(\mathcal{D})) + H(P(\mathcal{D}))}_{=0}\\
&=\mathbf{GAim1} - H(P(\phi(\mathbf{X}) \mid \psi(\mathbf{Y}), \mathcal{D})) - H(P(\mathcal{D})) + H(P(\psi(\mathbf{Y}), \mathcal{D})) + H(P(\phi(\mathbf{X}) \mid \mathcal{D})) + H(P(\mathcal{D}))\\
&=\mathbf{GAim1} - H(P(\phi(\mathbf{X}) \mid \psi(\mathbf{Y}))) + H(P(\phi(\mathbf{X})))\\
&=\mathbf{GAim1} + \mathbf{GReg2}.
\end{aligned}
$$

**Derivation details of Equation 8.** For a Gaussian distribution $\mathcal{N}(x; \mu, \Sigma)$ with $D$ dimension, its entropy is:

$$H(x) = - \int p(x) \ln p(x) dx \tag{22}$$

$$= - \int p(x)[\ln\left((2\pi)^{-\frac{D}{2}} |\Sigma|^{-\frac{1}{2}}\right) - \frac{1}{2}(x - \mu)^\top \Sigma^{-1}(x - \mu)]dx$$

$$= \ln\left((2\pi)^{\frac{D}{2}} |\Sigma|^{-\frac{1}{2}}\right) + \frac{1}{2} \int p(x)[(x - \mu)^\top \Sigma^{-1}(x - \mu)]dx$$

$$= \ln\left((2\pi)^{\frac{D}{2}} |\Sigma|^{-\frac{1}{2}}\right) + \frac{1}{2} \int p(y) \times y^\top dy$$

$$= \ln\left((2\pi)^{\frac{D}{2}} |\Sigma|^{-\frac{1}{2}}\right) + \frac{1}{2} \sum_{d=1}^{D} \mathbb{E}[y_d^2]$$

$$= \ln\left((2\pi)^{\frac{D}{2}} |\Sigma|^{-\frac{1}{2}}\right) + \frac{D}{2} \tag{23}$$

$$= \frac{D}{2}(1 + \ln(2\pi)) + \frac{1}{2} \ln |\Sigma|.$$

Then Equation 8 equals:

$$H(\mathcal{N}(\phi(\mathbf{X}); \mu_{\mathbf{X}}, \Sigma_{\mathbf{XX}})) - H(\mathcal{N}(\phi(\mathbf{X}) \mid \psi(\mathbf{Y}); \mu_{\mathbf{X}|\mathbf{Y}}, \Sigma_{\mathbf{XX}|\mathbf{Y}})) \tag{24}$$

$$= \frac{D}{2}(1 + \ln(2\pi)) + \frac{1}{2} \ln |\Sigma_{\mathbf{XX}}| - \frac{D}{2}(1 + \ln(2\pi)) - \frac{1}{2} \ln |\Sigma_{\mathbf{XX}|\mathbf{Y}}| \tag{25}$$

$$= \frac{1}{2} \ln(|\Sigma_{\mathbf{XX}}|) - \ln(|\Sigma_{\mathbf{XX}|\mathbf{Y}}|),$$

$$= \frac{1}{2} \ln(\frac{|\Sigma_{\mathbf{XX}}|}{|\Sigma_{\mathbf{XX}|\mathbf{Y}}|}).$$

**Empirical risk.** The empirical risk introduced by the whole model $\theta$ *w.r.t* $\mathbf{X}, \mathbf{Y}$ is determined by a convex loss function $L(\theta)$. Following (Perlaza et al., 2022), the empirical risk considering $\mathcal{O}$ is:

$$R(\theta) = \int L(\theta)dP(\theta) + H(\mathbb{E}[P(\phi(\mathbf{X}_n), \psi(\mathbf{Y}_n))]) + D_{\text{KL}}(\mathbb{E}[P(\phi(\mathbf{X}_n), \psi(\mathbf{Y}_n))] \| \mathcal{O}) \tag{26}$$
$$- H(P(\phi(\mathbf{X}) \mid \psi(\mathbf{Y}))) + H(P(\phi(\mathbf{X}))).$$

**Proof of Using $\psi$ *v.s.* not using $\psi$.** Using *Jensen's inequality*, due to $\mathbf{Y}, \psi(\mathbf{Y})$ contains the same amount of useful information as $\mathbf{Y}$, we have:

$$H(\mathbf{Y}) \geq H(\psi(\mathbf{Y})). \tag{27}$$

Therefore, we have

$$H(\mathbb{E}[P(\phi(\mathbf{X}_n), \mathbf{Y}_n)]) \tag{28}$$
$$= H(\mathbb{E}[P(\phi(\mathbf{X}_n))]) + H(\mathbb{E}[P(\mathbf{Y}_n \mid \phi(\mathbf{X}_n))])$$
$$\geq H(\mathbb{E}[P(\phi(\mathbf{X}_n))]) + H(\mathbb{E}[P(\psi(\mathbf{Y}_n) \mid \phi(\mathbf{X}_n))])$$
$$= H(\mathbb{E}[P(\phi(\mathbf{X}_n), \psi(\mathbf{Y}_n))]).$$

Therefore, for the risk of $\theta^{n\psi}$:

$$\sup R(\theta^{n\psi}) = \sup \min_{\phi} [\int L(\theta)dP(\theta) + H(\mathbb{E}[P(\phi(\mathbf{X}_n), \mathbf{Y}_n)])] + b, \tag{29}$$

and the risk of $\theta^{n\psi}$:

$$\sup R(\theta^{\psi}) = \sup \min_{\phi} [\int L(\theta)dP(\theta) + H(\mathbb{E}[P(\phi(\mathbf{X}_n), \psi(\mathbf{Y}_n))])] + b, \tag{30}$$

where $b \triangleq D_{\mathrm{KL}}(\mathbb{E}[P(\phi(\mathbf{X}_n), \psi(\mathbf{Y}_n))] \| \mathcal{O}) - H(P(\phi(\mathbf{X}) \mid \psi(\mathbf{Y}))) + H(P(\phi(\mathbf{X})))$, we have:

$$\sup R(\theta^{n\psi}) \geq \sup R(\theta^{\psi}). \tag{31}$$

**Proof of incorporating conditions leads to lower generalization risk on learning invariant representations.** For the risks of the model having parameters $\theta^c$ trained with using conditions, we have:

$$\sup R(\theta^c) = \sup \min_{\phi} [\int L(\theta) dP(\theta) + H(\mathbb{E}[P(\phi(\mathbf{X}_n), \psi(\mathbf{Y}_n))])] + b, \tag{32}$$

where $b \triangleq D_{\mathrm{KL}}(\mathbb{E}[P(\phi(\mathbf{X}_n), \psi(\mathbf{Y}_n))] \| \mathcal{O}) - H(P(\phi(\mathbf{X}) \mid \psi(\mathbf{Y}))) + H(P(\phi(\mathbf{X})))$. For $R(\theta^{nc})$ that trained without using conditions, it has:

$$\sup R(\theta^{nc}) = \sup \min_{\phi, \psi} [\int L(\theta) dP(\theta) + H(\mathbb{E}[P(\phi(\mathbf{X}_n))])] + H(\mathbb{E}[P(\phi(\mathbf{X}_n), \psi(\mathbf{Y}_n))]) \tag{33}$$
$$- H(\mathbb{E}[P(\phi(\mathbf{X}_n))]) + b$$
$$= \sup \min_{\phi, \psi} [\int L(\theta) dP(\theta) + H(\mathbb{E}[P(\phi(\mathbf{X}_n))])] + H(\mathbb{E}[P(\psi(\mathbf{Y}_n) \mid \phi(\mathbf{X}_n))]) + b.$$

Due to the inequality:

$$` \sup \min_{\phi} [\ H(\mathbb{E}[P(\phi(\mathbf{X}_n), \psi(\mathbf{Y}_n))])\ ] \tag{34}$$

$$= \sup \min_{\phi} [\ H(\mathbb{E}[P(\phi(\mathbf{X}_n))]) + H(\mathbb{E}[P(\psi(\mathbf{Y}_n) \mid \phi(\mathbf{X}_n))])\ ] \tag{35}$$

$$\leq \sup \min_{\phi, \psi} [\ H(\mathbb{E}[P(\phi(\mathbf{X}_n))])\ ] + H(\mathbb{E}[P(\psi(\mathbf{Y}_n) \mid \phi(\mathbf{X}_n))]), \tag{36}$$

we have

$$\sup R(\theta^{nc}) \geq \sup R(\theta^c). \tag{37}$$

## B  OBJECTIVE DERIVATION DETAILS OF MANY PREVIOUS METHODS.

This section shows how we uniformly simplify the objectives of previous methods.

**ERM (Gulrajani & Lopez-Paz, 2020): The basic method.** The basic method does not focus on minimizing GJSD. Therefore, there are no terms for **Aim 1**. For **Aim 2** it directly minimize $H(P(\phi(\mathbf{X}), \mathbf{Y}))$.

**DANN (Ganin et al., 2016): Minimize feature divergences of source domains.** DANN (Ganin et al., 2016) minimizes feature divergences of source domains adverbially without considering conditions. Therefore its empirical objective for **Aim 1** is

$$\min_{\phi} H(\mathbb{E}[P(\phi(\mathbf{X}_n))]) - a \tag{38}$$

For **Aim 2** it directly minimizes $H(P(\phi(\mathbf{X}), \mathbf{Y}))$.

**CORAL Sun & Saenko (2016): Minimize the distance between the second-order statistics of source domains.** Since CORAL Sun & Saenko (2016) only minimizes the second-order distance between souce feature distributions, its objective can be summarized as:

$$\min_{\phi} H(P(\phi(\mathbf{X}), \mathbf{Y})) + H(P(\phi(\mathbf{X}))) - H(\mathbb{E}[P(\phi(\mathbf{X}_n))]). \tag{39}$$

By grouping it, CORAL (Sun & Saenko, 2016) has $-H(\mathbb{E}[P(\phi(\mathbf{X}_n))])$ for **Aim 1** and $H(P(\phi(\mathbf{X}), \mathbf{Y})) + H(P(\phi(\mathbf{X})))$ for **Aim 2**.

**CIDG (Li et al., 2018c): Minimizing the conditioned domain gap.** CIDG (Li et al., 2018c) tries to learn conditional domain invariant features:

$$\min_{\phi} H(\mathbb{E}[P(\phi(\mathbf{X}_n), \mathbf{Y}_n)]). \tag{40}$$

For **Aim 2** it directly minimizes $H(P(\phi(\mathbf{X}), \mathbf{Y}))$.

**MDA (Hu et al., 2020): Minimizing domain gap compared to the decision gap.** Some previous works, such as MDA (Hu et al., 2020), follow the hypothesis that the generalization is guaranteed while the decision gap is larger than the domain gap. Therefore, instead of directly minimizing the domain gap, MDA minimizes the ratio between the domain gap and the decision gap. The overall objective of MDA can be summarized as:

$$
\min_{\phi} H(P(\phi(\mathbf{X}), \mathbf{Y})) + H(P(\phi(\mathbf{X}))) \tag{41}
$$
$$
+ (H(\mathbb{E}[P(\phi(\mathbf{X}_n), \mathbf{Y}_n)]) - \underbrace{\mathbb{E}[H(P(\phi(\mathbf{X}_n), \mathbf{Y}_n))]}_{constant})
$$
$$
- (H(\mathbb{E}[P(\phi(\mathbf{X}_n) \mid \mathbf{Y}_n)]) - \underbrace{\mathbb{E}[H(P(\phi(\mathbf{X}) \mid \mathbf{Y}))]}_{constant}) + \underbrace{\mathbb{E}[H(P(\phi(\mathbf{X}), \mathbf{Y}))]}_{constant}.
$$

Since the entropy is non-negative and the constants can be omitted, Equation 41 is equivalent to:

$$
\min_{\phi} H(P(\phi(\mathbf{X}), \mathbf{Y})) + H(P(\phi(\mathbf{X}))) + H(\mathbb{E}[P(\phi(\mathbf{X}_n), \mathbf{Y}_n)]) - H(\mathbb{E}[P(\phi(\mathbf{X}) \mid \mathbf{Y})])) + a.
$$
$$\tag{42}$$

By grouping Equation 42, we have that for **Aim 1** it minimizes $\min_{\phi} H(\mathbb{E}[P(\phi(\mathbf{X}_n), \mathbf{Y}_n)])$, and for **Aim 2** it minimizes $H(P(\phi(\mathbf{X}), \mathbf{Y})) - H(\mathbb{E}[P(\phi(\mathbf{X}) \mid \mathbf{Y})])) + H(P(\phi(\mathbf{X})))$.

**MIRO (Junbum et al., 2022), SIMPLE (Li et al., 2022): Using pre-trained models as $\mathcal{O}$.** One feasible way to obtain $\mathcal{O}$ is adopting pre-trained oracle models such as MIRO (Junbum et al., 2022) and SIMPLE (Li et al., 2022). Note that the pre-trained models are exposed to additional data besides those provided. Therefore, for **Aim 1**: they have:

$$
\min_{\phi} D_{\mathrm{KL}}(P(\phi(X)|Y)\|\mathcal{O}) - \underbrace{\mathbb{E}[H(P(\phi(\mathbf{X}_n), \mathbf{Y}_n))]}_{constant}.
$$

Differently, MIRO only uses one pre-trained model, as its $\mathcal{O} \triangleq \mathcal{O}^1$; meanwhile, SIMPLE combines $K$ pre-trained models as the oracle model: $\mathcal{O} \triangleq \sum_{k=1}^{K} v_k \mathcal{O}^k$ where $v$ is the weight vector. For **Aim 2** it directly minimizes $H(P(\phi(\mathbf{X}), \mathbf{Y}))$.

**RobustNet (Choi et al., 2021).** RobustNet employs the instance selective whitening loss, which disentangles domain-specific and domain-invariant properties from higher-order statistics of the feature representation and selectively suppresses domain-specific ones. Therefore, it implicitly whitens the $\mathbf{Y}$-irrelevant features in $\mathbf{X}$. Thus, its objective can be simplified as:

$$
\min_{\phi,\psi} H(P(\phi(\mathbf{X}), \psi(\mathbf{Y}))) - H(P(\phi(\mathbf{X}) \mid \psi(\mathbf{Y}))) + H(P(\phi(\mathbf{X}))). \tag{43}
$$

## B.1 OTHER FINDINGS

**What makes a better $\mathcal{O}$.** As demonstrated in Equation 6, $\mathcal{O}$ plays a crucial role in PUB by anchoring a space where the relationship between $\mathbf{X}$ and $\mathbf{Y}$ is preserved. Ideally, having one $\mathcal{O}$ that provides general representations for all seen and unseen domains leads to the best results, one finding supported by MIRO and SIMPLE. However, even though SIMPLE++ combines 283 pre-trained models, achieving the 'perfect' $\mathcal{O}$ remains unattained. Therefore, this paper primarily focuses on discussing how our proposed objectives can improve the model performance when a fixed $\mathcal{O}$ is provided.

**Comparison with MDA: Minimizing domain gap compared to the decision gap.** MDA (Hu et al., 2020), guided by the hypothesis "guaranteed generalization only when the decision gap exceeds the domain gap", aims to minimize the ratio between the domain gap and the decision gap. This approach facilitates learning $\mathcal{D}$-independent conditional features, enhancing class separability across domains. As Table 1 illustrates, MDA's **Reg2** objective can also be interpreted as suppressing invalid causality, aligning with our approach. However, MDA's implementation requires manual selection of $\phi(\mathbf{X})$ from the same $\mathbf{Y}$ without using $\psi$ and **GReg2**. Our method further relaxes MDA's assumption, extending the application of the objective and making it also applicable to tasks besides classification, such as segmentation.

Table 9: Synthetic data details for each experiment.

| | | |
|---|---|---|
| $hx$ | $hx \sim \mathcal{N}(hx; 0, 1)$ | |
| $hy$ | $hy = hx$ | |
| Data 1 | Without distribution shift | With affine transformations |
| $X_1$ | $x_1^1 = hx$ | $x_1^2 = x_1^1 + \epsilon \sim \mathcal{N}(\epsilon; 0, 0.3)$ |
| $Y_1$ | $y_1^1 = hy$ | $y_1^2 = y_1^1 + \epsilon \sim \mathcal{N}(\epsilon; 0, 0.3)$ |
| $X_2$ | $x_2^1 = hx$ | $x_2^2 = 4 \times x_2^1 + \epsilon \sim \mathcal{N}(\epsilon; 0.5, 0.3)$ |
| $Y_2$ | $y_2^1 = hy$ | $y_2^2 = 4 \times y_2^1 + 0.3$ |
| $X_3$ | $x_3^1 = hx$ | $x_3^2 = 2 \times x_3^1 + \epsilon \sim \mathcal{N}(\epsilon; -0.3, 0.2)$ |
| $Y_3$ | $y_3^1 = hy$ | $y_3^2 = 0.5 \times y_3^1 - 0.2$ |
| Data 2 | With distribution shift | With affine transformations |
| $X_1$ | $x_1^1 = hx$ | $x_1^2 = x_1^1 + \epsilon \sim \mathcal{N}(\epsilon; 0, 0.3)$ |
| $Y_1$ | $y_1^1 = hy$ | $y_1^2 = y_1^1 + \epsilon \sim \mathcal{N}(\epsilon; 0, 0.3)$ |
| $X_2$ | $x_2^1 = hx + \epsilon \sim \mathcal{N}(\epsilon; -0.1, 0.1)$ | $x_2^2 = 4 \times x_2^1 + \epsilon \sim \mathcal{N}(\epsilon; 0.3, 0.3)$ |
| $Y_2$ | $y_2^1 = hy + \epsilon \sim \mathcal{N}(\epsilon; 0.2, 0.1)$ | $y_2^2 = 8 \times y_2^1 - 0.3$ |
| $X_3$ | $x_3^1 = hx + \epsilon \sim \mathcal{N}(\epsilon; 0.4, 0.2)$ | $x_3^2 = -1 \times x_3^1 + \epsilon \sim \mathcal{N}(\epsilon; -0.3, 0.2)$ |
| $Y_3$ | $y_3^1 = hy + \epsilon \sim \mathcal{N}(\epsilon; -0.4, 0.2)$ | $y_3^2 = \epsilon \sim \mathcal{N}(\epsilon; 0, 0.2)$ |
| Data 3 | Without distribution shift | With squared, cubed transformations or noises |
| $X_1$ | $x_1^1 = hx$ | $x_1^2 = x_1^1 + \epsilon \sim \mathcal{N}(\epsilon; 0, 0.3)$ |
| $Y_1$ | $y_1^1 = hy$ | $y_1^2 = y_1^1 + \epsilon \sim \mathcal{N}(\epsilon; 0, 0.3)$ |
| $X_2$ | $x_2^1 = hx$ | $x_2^2 = 4 \times x_2^1 ** 3 + \epsilon \sim \mathcal{N}(\epsilon; 0.5, 0.3)$ |
| $Y_2$ | $y_2^1 = hy$ | $y_2^2 = 4 \times y_2^1 ** 2 + 0.3$ |
| $X_3$ | $x_3^1 = hx$ | $x_3^2 = 2 \times x_3^1 ** 2 + \epsilon \sim \mathcal{N}(\epsilon; -0.3, 0.2)$ |
| $Y_3$ | $y_3^1 = hy$ | $y_3^2 = 0.5 \times y_3^1 ** 3 - 0.2$ |
| Data 4 | With distribution shift | With squared, cubed transformations or noises |
| $X_1$ | $x_1^1 = hx$ | $x_1^2 = x_1^1 + \epsilon \sim \mathcal{N}(\epsilon; 0, 0.3)$ |
| $Y_1$ | $y_1^1 = hy$ | $y_1^2 = y_1^1 + \epsilon \sim \mathcal{N}(\epsilon; 0, 0.3)$ |
| $X_2$ | $x_2^1 = hx + \epsilon \sim \mathcal{N}(\epsilon; -0.1, 0.1)$ | $x_2^2 = 4 \times x_2^1 ** 3 + \epsilon \sim \mathcal{N}(\epsilon; 0.5, 0.3)$ |
| $Y_2$ | $y_2^1 = hy + \epsilon \sim \mathcal{N}(\epsilon; 0.2, 0.1)$ | $y_2^2 = 4 \times y_2^1 ** 2 + 0.3$ |
| $X_3$ | $x_3^1 = hx + \epsilon \sim \mathcal{N}(\epsilon; 0.4, 0.2)$ | $x_3^2 = 2 \times x_3^1 ** 2 + \epsilon \sim \mathcal{N}(\epsilon; -0.3, 0.2)$ |
| $Y_3$ | $y_3^1 = hy + \epsilon \sim \mathcal{N}(\epsilon; -0.4, 0.2)$ | $y_3^2 = 0.5 \times y_3^1 ** 3 - 0.2$ |

## C  EXPERIMENTAL DETAILS AND PARAMETERS

Note that we set $v_{A2} = 1$ for all experiments. More details can be found in supplementary materials which contain codes of the implementations.

**Synthetic experimental details.** The latent features in all three domains are added some distributional shifts and used as the first group in the raw features (denoted as $x_n^1, y_n^1$ where $n \in 1, 2, 3$ represent which domain it belongs to). Then some domain-conditioned transformations are applied to shifted features, or some pure random noises are used as the second group in the raw features (denoted as $x_n^2, y_n^2$). Therefore the constructed $X_{n \in \{1,2,3\}} = [x_n^1, x_n^2], Y_{n \in \{1,2,3\}} = [y_n^1, y_n^2]$ both contain features that dependents on $D$. Details of each synthetic data are exhibited in Table 9. We generate 10000 samples for training and 100 samples for validation and testing. For $\phi, \psi$, we use three-layer MLP and one linear layer for regression prediction. All experiments are conducted with $v_{A1}, v_{R1}, v_{R2} = 0.1$.

**Segmentation experimental details.** We follow the experimental settings of RobustNet for segmentation experiments. Specifically, we use all hyper-parameters used by RobustNet and set $v_{A1} = 0.0001, v_{R1} = 0.0001$.

**Main experimental details.** We list the hyper-parameters in Table 10 to reproduce our results.

**Ablation studies experimental details.** We run each experiment in three trials with seeds: $[0, 1, 2]$. We use SWAD for all ablation studies to alleviate the effeteness of hyper-parameters. All ablation studies share the same hyper-parameters but add different combinations of terms. Full settings are

Table 10: Parameter settings of classification tasks. Notations: WD: weight decay; TR: tolerance ratio; CF: checkpoint freq. - denotes that for where the default settings are used.

| Use ResNet-50 without SWAD | $v2$ | $v3$ | $v1$ | lr mult | lr | dropout | WD | TR | CF |
|---|---|---|---|---|---|---|---|---|---|
| TerraIncognita | 0.1 | 0.1 | 0.2 | 12.5 | - | - | - | - | - |
| OfficeHome | 0.1 | 0.001 | 0.1 | 20.0 | 3e-5 | 0.1 | 1e-6 | - | - |
| VLCS | 0.01 | 0.001 | 0.1 | 10.0 | 1e-5 | - | 1e-6 | 0.2 | 50 |
| PACS | 0.01 | 0.01 | 0.01 | 25.0 | - | - | - | - | - |
| DomainNet | 0.1 | 0.1 | 0.1 | 7.5 | - | - | - | - | 500 |

| Use ResNet-50 with SWAD | $v2$ | $v3$ | $v1$ | lr mult | CF |
|---|---|---|---|---|---|
| TerraIncognita | 0.1 | 0.001 | 0.01 | 10.0 | - |
| OfficeHome | 0.1 | 0.1 | 0.3 | 10.0 | - |
| VLCS | 0.01 | 0.001 | 0.1 | 10.0 | 50 |
| PACS | 0.01 | 0.001 | 0.1 | 20.0 | - |
| DomainNet | 0.1 | 0.1 | 0.1 | 7.5 | 500 |

| Use RegNetY-16GF with and without SWAD | $v2$ | $v3$ | $v1$ | lr mult | CF |
|---|---|---|---|---|---|
| TerraIncognita | 0.01 | 0.01 | 0.01 | 2.5 | - |
| OfficeHome | 0.01 | 0.1 | 0.1 | 0.1 | - |
| VLCS | 0.01 | 0.01 | 0.1 | 2.0 | 50 |
| PACS | 0.01 | 0.1 | 0.1 | 0.1 | - |
| DomainNet | 0.1 | 0.1 | 0.1 | 7.5 | 500 |

Table 11: Parameter settings of ablation studies. Notations: WD: CF: checkpoint freq. - denotes that for where the default settings are used.

| Ablation studies on OfficeHome | $v2$ | $v3$ | $v1$ | lr mult | use **iAim1** | use **iT3** |
|---|---|---|---|---|---|---|
| Base (ERM) | 0.0 | 0.0 | 0.0 | 0.1 | False | False |
| Base +iAim1 (DANN) | 0.0 | 0.0 | 0.1 | 0.1 | True | False |
| Base + GAim1 (CDANN, CIDG) | 0.0 | 0.0 | 0.1 | 0.1 | False | False |
| Base +iReg2 (CORAL+$\psi$) | 0.0 | 0.1 | 0.0 | 0.1 | False | True |
| Base + GReg2 | 0.0 | 0.1 | 0.0 | 0.1 | False | False |
| Base + GAim1 + GReg2 (MDA+$\psi$) | 0.0 | 0.1 | 0.1 | 0.1 | False | False |
| Base + GReg1 (MIRO, SIMPLE) | 0.01 | 0.0 | 0.0 | 0.1 | False | False |
| Base + GReg1 +iAim1 | 0.01 | 0.0 | 0.1 | 0.1 | False | True |
| Base + GReg1 + GAim1 | 0.01 | 0.0 | 0.1 | 0.1 | False | False |
| Base + GReg1 +iReg2 | 0.01 | 0.1 | 0.0 | 0.1 | True | False |
| Base + GReg1 + GReg2 | 0.01 | 0.1 | 0.0 | 0.1 | False | False |
| Base + GReg1 + GAim1 + GReg2 (Ours) | 0.01 | 0.1 | 0.1 | 0.1 | False | False |

reported in Table 11. Especially, CORAL's (Sun & Saenko, 2016) objective focuses on minimizing the learned feature covariance discrepancy between source and target, requiring target data access and only regards second-order statistics. For a fair comparison, we adapt its approach to minimize feature covariances across seen domains.

# D  MORE RESULTS

**Segmentation results.** The segmentation result visualization is displayed in Figure 4.

**Classification results.** We show the results of each category for the classification experiments as Table 13.

RobustNet GT Ours

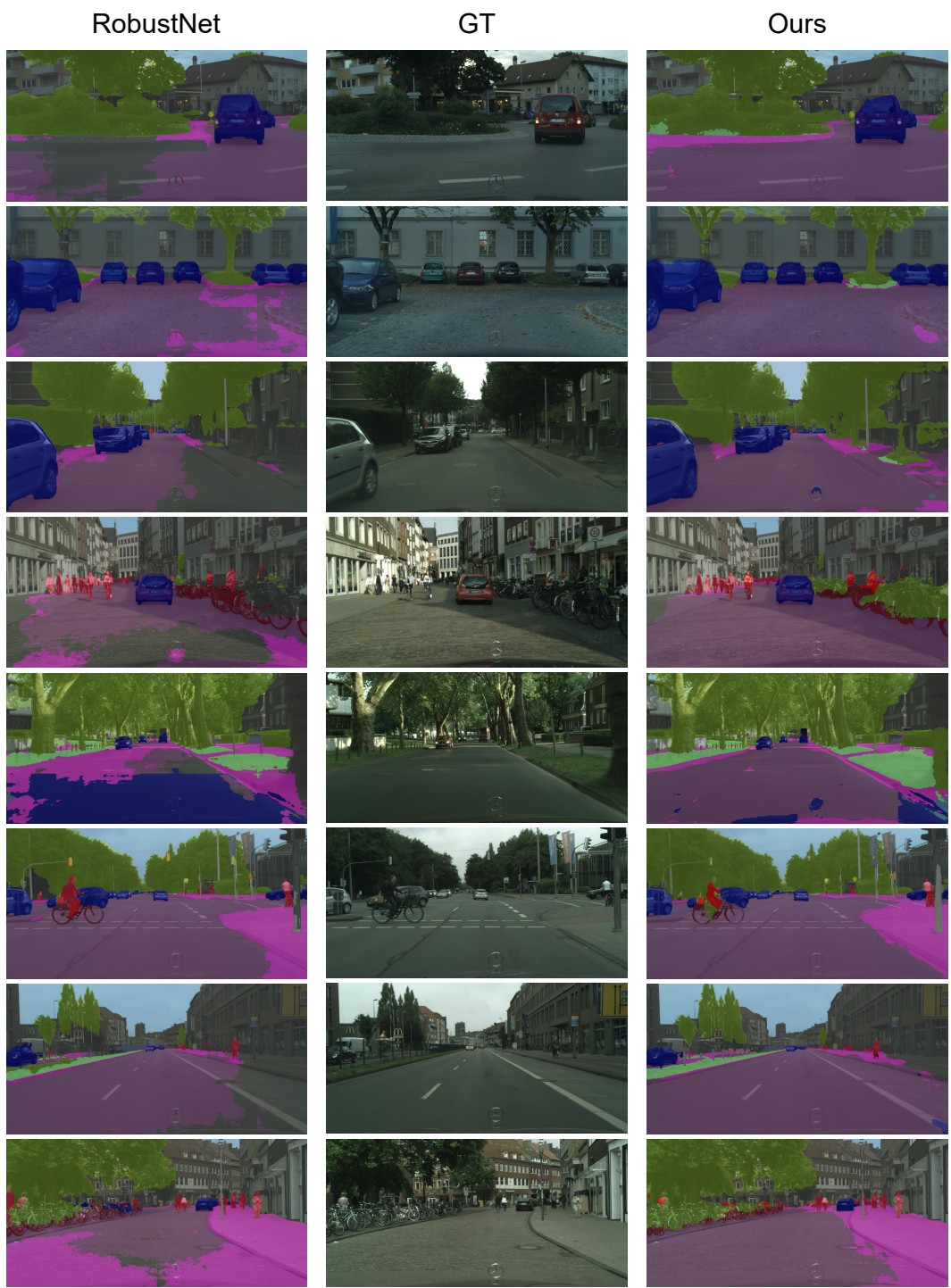

Figure 4: Segmentation results visualization between RobostNet and ours on Cityscapes.

Table 12: Main results: the proposed method compared with previous non-ensemble and ensemble mDG methods. The best results for each group are highlighted in **bold**.

| | **Non-ensemble methods** | | | | | |
| | **PACS** | **VLCS** | **OfficeHome** | **TerraInc** | **DomainNet** | **Avg.** |
|---|---|---|---|---|---|---|
| MMD (Li et al., 2018b) | 84.7±0.5 | 77.5±0.9 | 66.3±0.1 | 42.2±1.6 | 23.4±9.5 | 58.8 |
| Mixstyle (Zhou et al., 2021) | 85.2±0.3 | 77.9±0.5 | 60.4±0.3 | 44.0±0.7 | 34.0±0.1 | 60.3 |
| GroupDRO (Sagawa et al., 2019) | 84.4±0.8 | 76.7±0.6 | 66.0±0.7 | 43.2±1.1 | 33.3±0.2 | 60.7 |
| IRM Arjovsky et al. (2019) | 83.5±0.8 | 78.5±0.5 | 64.3±2.2 | 47.6±0.8 | 33.9±2.8 | 61.6 |
| ARM (Zhang et al., 2021) | 85.1±0.4 | 77.6±0.3 | 64.8±0.3 | 45.5±0.3 | 35.5±0.2 | 61.7 |
| VREx (Krueger et al., 2021) | 84.9±0.6 | 78.3±0.2 | 66.4±0.6 | 46.4±0.6 | 33.6±2.9 | 61.9 |
| CDANN (Li et al., 2018d) | 82.6±0.9 | 77.5±0.1 | 65.8±1.3 | 45.8±1.6 | 38.3±0.3 | 62.0 |
| DANN (Ganin et al., 2016) | 83.6±0.4 | 78.6±0.4 | 65.9±0.6 | 46.7±0.5 | 38.3±0.1 | 62.6 |
| RSC (Huang et al., 2020) | 85.2±0.9 | 77.1±0.5 | 65.5±0.9 | 46.6±1.0 | 38.9±0.5 | 62.7 |
| MTL (Blanchard et al., 2021) | 84.6±0.5 | 77.2±0.4 | 66.4±0.5 | 45.6±1.2 | 40.6±0.1 | 62.9 |
| MLDG (Li et al., 2018a) | 84.9±1.0 | 77.2±0.4 | 66.8±0.6 | 47.7±0.9 | 41.2±0.1 | 63.6 |
| Fish (Shi et al., 2021) | 85.5±0.3 | 77.8±0.3 | 68.6±0.4 | 45.1±1.3 | 42.7±0.2 | 63.9 |
| ERM (Vapnik, 1998) | 84.2±0.1 | 77.3±0.1 | 67.6±0.2 | 47.8±0.6 | 44.0±0.1 | 64.2 |
| SagNet (Nam et al., 2021) | **86.3**±0.2 | 77.8±0.5 | 68.1±0.1 | 48.6±1.0 | 40.3±0.1 | 64.2 |
| SelfReg (Kim et al., 2021) | 85.6±0.4 | 77.8±0.9 | 67.9±0.7 | 47.0±0.3 | 42.8±0.0 | 64.2 |
| CORAL (Sun & Saenko, 2016) | 86.2±0.3 | 78.8±0.6 | 68.7±0.3 | 47.6±1.0 | 41.5±0.1 | 64.5 |
| mDSDI Bui et al. (2021) | 86.2±0.2 | 79.0±0.3 | 69.2±0.4 | 48.1±1.4 | 42.8±0.1 | 65.1 |
| | Use ResNet-50 (He et al., 2016) as oracle model. | | | | | |
| MIRO (Junbum et al., 2022) | 85.4±0.4 | 79.0±0.3 | 70.5±0.4 | 50.4±1.1 | 44.3±0.2 | 65.9 |
| **Ours** | 85.6±0.3 | **79.2**±0.3 | **70.7**±0.2 | **51.1**±0.9 | **44.6**±0.1 | **66.3** |
| | Use RegNetY-16GF (Singh et al., 2022) as oracle model. | | | | | |
| MIRO | **97.4**±0.2 | 79.9±0.6 | 80.4±0.2 | 58.9±1.3 | 53.8±0.1 | 74.1 |
| **Ours** | 97.3±0.1 | **82.4**±0.6 | **80.8**±0.6 | **60.7**±1.8 | **54.6**±0.1 | **75.1** |
| | **Ensemble methods** | | | | | |
| | **PACS** | **VLCS** | **OfficeHome** | **TerraInc** | **DomainNet** | **Avg.** |
| | Use multiple oracle models. | | | | | |
| SIMPLE (Li et al., 2022) | 88.6±0.4 | 79.9±0.5 | 84.6±0.5 | 57.6±0.8 | 49.2±1.1 | 72.0 |
| SIMPLE++ (Li et al., 2022) | **99.0**±0.1 | **82.7**±0.4 | **87.7**±0.4 | **59.0**±0.6 | **61.9**±0.5 | **78.1** |
| | Use ResNet-50 (He et al., 2016) as oracle model. | | | | | |
| MIRO + SWAD | **88.4**±0.1 | 79.6±0.2 | 72.4±0.1 | 52.9±0.2 | 47.0±0.0 | 68.1 |
| **Ours + SWAD** | **88.4**±0.1 | **79.6**±0.1 | **72.5**±0.2 | **53.0**±0.7 | **47.3**±0.1 | **68.2** |
| | Use RegNetY-16GF (Singh et al., 2022) as oracle model. | | | | | |
| MIRO + SWAD | 96.8±0.2 | 81.7±0.1 | 83.3±0.1 | 64.3±0.3 | 60.7±0.0 | 77.3 |
| **Ours + SWAD** | **97.9**±0.3 | **82.2**±0.3 | **84.7**±0.2 | **65.0**±0.2 | **61.3**±0.2 | **78.2** |

Table 13: More results of our method for each category in all datasets.

| TerraIncognita | Location 100 | Location 38 | Location 43 | Location 46 | Avg. |
|---|---|---|---|---|---|
| ResNet-50 | 59.8±1.0 | 45.3±1.7 | 57.1±1.8 | 38.2±5 | 50.1±1.2 |
| + SWAD | 61.2±1.4 | 48.4±1.6 | 60.0±0.4 | 42.5±1.1 | 53.0±0.7 |
| RegNetY-16GF | 73.3±3.3 | 54.7±1.4 | 67.1±0.3 | 48.6±6.5 | 60.7±1.8 |
| + SWAD | 74.3±1.5 | 59.2±1.2 | 70.6±1.1 | 56.0±0.8 | 65.0±0.2 |

| OfficeHome | art | clipart | product | real | Avg. |
|---|---|---|---|---|---|
| ResNet-50 | 68.9±0.3 | 56.2±1.7 | 79.9±0.6 | 82.0±0.4 | 70.7±0.2 |
| + SWAD | 68.9±0.6 | 58.2±0.6 | 80.4±0.3 | 82.6±0.4 | 72.5±0.2 |
| RegNetY-16GF | 79.7±1.6 | 67.7±1.8 | 87.8±0.8 | 87.9±0.7 | 80.8±0.6 |
| + SWAD | 84.1±0.2 | 74.3±0.9 | 89.9±0.4 | 90.6±0.1 | 84.7±0.2 |

| VLCS | caltech101 | labelme | sun09 | voc2007 | Avg. |
|---|---|---|---|---|---|
| ResNet-50 | 98.3±0.4 | 65.9±1 | 73.4±0.8 | 79.3±1.3 | 79.2±0.3 |
| + SWAD | 98.9±0.4 | 63.6±0.2 | 76.4±0.5 | 79.5±0.6 | 79.6±0.1 |
| RegNetY-16GF | 97.9±1.3 | 66.8±2.1 | 80.8±1 | 83.9±1.8 | 82.4±0.6 |
| + SWAD | 98.4±0.1 | 65.5±1.4 | 79.9±0.4 | 84.9±0.9 | 82.2±0.3 |

| PACS | art_painting | cartoon | photo | sketch | Avg. |
|---|---|---|---|---|---|
| ResNet-50 | 90.1±0.6 | 83.9±0.2 | 97.6±0.5 | 82.3±0.7 | 88.4±0.1 |
| + SWAD | 84.7±1.0 | 81.7±2.4 | 97.5±0.4 | 80.5±1.8 | 85.6±0.3 |
| RegNetY-16GF | 97.5±1.0 | 97.0±0.2 | 99.4±0.2 | 95.2±0.4 | 97.3±0.1 |
| + SWAD | 98.3±0.3 | 98.0±0.1 | 99.5±0.3 | 95.3±0.8 | 97.9±0.0 |

| DomainNet | clipart | info | painting | quickdraw | real | sketch | Avg. |
|---|---|---|---|---|---|---|---|
| ResNet-50 | 63.4±0.3 | 22.4±0.4 | 51.4±0.4 | 13.4±0.8 | 64.4±0.3 | 52.4±0.4 | 44.6±0.1 |
| + SWAD | 66.4±0.3 | 23.8±0.1 | 54.5±0.3 | 15.8±0.1 | 67.5±0.1 | 55.8±0.0 | 47.3±0.1 |
| RegNetY-16GF | 74.0±0.3 | 39.5±1.5 | 61.5±0.3 | 16.3±1.2 | 73.9±1.5 | 62.8±2.4 | 54.6±0.1 |
| + SWAD | 79.0±0.0 | 46.9±0.4 | 69.9±0.4 | 20.7±0.6 | 81.1±0.3 | 70.3±0.4 | 61.3±0.2 |

