# OpenReview forum: "Rethinking Multi-domain Generalization with A General Learning Objective"
_ICLR.cc/2024/Conference — ICLR 2024 Conference Withdrawn Submission_

### Official Review · Reviewer_YmGa · 2023-10-23

**Soundness:** 3 good
**Presentation:** 3 good
**Contribution:** 3 good
**Rating:** 6
**Confidence:** 3

**Summary:**

Overall, the paper presents a novel approach to the multi-domain generalization (mDG) problem and introduces a general learning objective that addresses certain limitations of existing methods.

**Strengths:**

The paper offers a novel perspective on multi-domain generalization by introducing a general learning objective. The idea of relaxing the static distribution assumption of Y through a learnable mapping ψ is a significant departure from traditional mDG approaches. The paper also demonstrates how this objective can be applied to different mDG tasks, including regression, segmentation, and classification. The theoretical analysis provides strong arguments for the proposed approach.

Technical Soundness: The paper appears technically sound, with a well-documented formulation of the proposed objective and a clear connection to previous mDG methods. The authors provide mathematical derivations, empirical results, and ablation studies to support their claims.

Clarity: The paper is relatively well-organized and clearly written. It provides mathematical details and explanations for key concepts, which makes it accessible to researchers in the field.

**Weaknesses:**

Complexity: The paper's content, including mathematical derivations and technical details, might be challenging for non-experts or those new to the field. While it's essential to provide in-depth technical content, the authors should consider adding more intuitive explanations to help a broader audience understand the key concepts.

Clarity: While the paper is mostly well-structured and clearly written, there are areas where the text could be made more reader-friendly. The use of jargon, equations, and acronyms can make it difficult to follow at times.

Visual Aids: The paper could benefit from more visual aids, such as diagrams or graphs, to help illustrate key points and make the content more accessible. Visual representations can be particularly helpful when explaining complex mathematical concepts.

**Questions:**

The DG baselines are old, where the latest baseline is CDANN from 2018.

---

### Official Review · Reviewer_tWvp · 2023-10-30

**Soundness:** 2 fair
**Presentation:** 2 fair
**Contribution:** 2 fair
**Rating:** 3
**Confidence:** 3

**Summary:**

This paper maps labels Y into a latent space to relax the Y-invariant constraint, and proposes a general learning objective that consist four parts to analyze and further improve the previous DG methods. Specifically, the four components of the proposed objective focus on learning domain invariance, maximizing a posterior, integrating prior and suppressing invalid causality based on the relaxed constraint respectively. Theoretical analysis and experiments on multiple tasks demonstrate the effectiveness and rationality of this method.

**Strengths:**

1.This work relaxes the static distribution assumption of Y by mapping it into latent space, which is conducive to learning domain invariance.

2.This work proposes a general learning objective for domain generalization that can be applied to various tasks and yield better generalization performance.

3.This work demonstrates that some previous methods actually optimize partially the proposed objective, proving its versatility and superiority.

**Weaknesses:**

1.The proposed objective seems to be an integration of multiple existing methods to some extent, with each component of the objective being able to find prototype in previous works, even including the relaxation technique. Therefore, I challenge the contribution of this work.

2.This paper claims that the proposed learning objective is general and can explain most of the  previous methods. However, I believe it should be “most of the previous methods that focused on learning invariant representations”. Many other groups of methods, which are based on causality, meta-learning, regularization, etc., may not be able to analyze through the proposed objective, limiting the universality of this work.

3.The baseline methods compared in experiments are not sufficient and lack relatively new works.

**Questions:**

1.What is the significance of suppressing the invalid causality from Y to X? The improvement of GReg2 seems trivial in some comparisons in Table 7.

2.What about the hyperparameter sensitivity of the proposed objective since there are four weights? Besides, I suggest the authors provide the training procedure for clarity.

3.There exist some typos, e.g., the inconsistency of GAim2 in Eq.3 and Eq. 10. The authors should carefully check the article.

---

### Official Review · Reviewer_Mt83 · 2023-11-05

**Soundness:** 3 good
**Presentation:** 2 fair
**Contribution:** 2 fair
**Rating:** 5
**Confidence:** 3

**Summary:**

This paper proposed a general learning objective for multi-source domain generalization, which includes four terms:
an empirical cross-entropy loss and a domain-invariant representation learning term, plus two regularization terms:
Reg1 aims to incorporate information from an oracle model, whereas Reg2 aims to suppress invalid causality.

In terms of implementation, authors introduce a new mapping function $\psi$ to project $y$ into a new space to alleviate label shift. Authors also assume both $\phi(X)$ and $\psi(Y)$ follow a multivariate Gaussian distribution, which reduces the difficulty on the closed-form expression of each term.

**Strengths:**

1. Learning a mapping function from $y$ to $\psi(y)$ to alleviate the distributional shift of $p(y)$ is new to me.
2. Using the generalized Jensen-Shannon divergence (GJSD) to derive objectives on learning domain-invariant representation and incorporating prior knowledge is also new to me.
3. It is good to see evaluations on both benchmark DomainBed and a visual semantic segmentation task.

**Weaknesses:**

1. It is always nice to propose a general objective for the problem of domain generalization. However, I have two concerns:

(1) The physical meaning of each term is not new, it is widely known that to improve generalization, we need to learn domain-invariant representation that also follows some causal constraints; recent studies also emphasized the role of incorporating prior information from oracle model or a pool of pre-trained models. In this sense, even though Eq.~(3) is a high-level summarization, it does not contribute new insights to the problem of DG.

(2) Table 1 is nice. However, it is hard to convince me that it summarizes most ideas from existing DG paper. At least, some recent papers are not included, which also aim to learn invariant representation from a causal perspective, or also incorproate some information-theoretic regularizatioin terms:

[1] Arjovsky, Martin, et al. "Invariant risk minimization." arXiv preprint arXiv:1907.02893 (2019).

[2] Ahuja, Kartik, et al. "Invariance principle meets information bottleneck for out-of-distribution generalization." Advances in Neural Information Processing Systems 34 (2021): 3438-3450.

[3] Zhao, Shanshan, et al. "Domain generalization via entropy regularization." Advances in Neural Information Processing Systems 33 (2020): 16096-16107.

[4] Liu, Xiaofeng, et al. "Domain generalization under conditional and label shifts via variational bayesian inference." arXiv preprint arXiv:2107.10931 (2021).

2. Some points are not clear to me:

(1) Eq.(7) is a bit unclear to me. It seems that the objective of GAim1 is $\min H(P(\phi(X),\psi(Y)|D))$ (according to your Table 1),
then does Eq.~(7) aim to say GAim1 <= GAim1 + GReg2?

(2) The GReg2 is simple and novel to me. Could you please explain more why minimizing the shift between unconditional and conditional features can be used to mitigate invalid causality?

(3) Why the term $a$ in Eq.~(5) is a constant w.r.t to $\phi$ and $\psi$? If we change either $\phi$ or $\psi$, the value of $a$ will change as well?

3. The results in Table 6 only contain two state-of-the-art methods. It is highly recommended to incorporate more results,
at least IRM, IRM+IB, etc. Otherwise, it is hard to judge, among the four terms, which term contributes more. It is also hard
to judge the performance improvement of the new method.

**Questions:**

1. Table 1 under the category of Aim2, it should be $H_c$, rather than $H$?
2. I was wondering how your general objective includes the ideas of [1-4] as special cases.
3. please refer to weaknesses point 2

---

### Official Review · Reviewer_rMjP · 2023-11-08

**Soundness:** 3 good
**Presentation:** 3 good
**Contribution:** 3 good
**Rating:** 5
**Confidence:** 3

**Summary:**

This paper reconsiders the learning objective for multi-domain generalization (mDG) and introduces a new, comprehensive learning objective suitable for interpreting and analyzing most existing mDG models. This general objective comprises two interrelated aims: acquiring domain-independent conditional features and maximizing a posterior. The exploration extends to include two effective regularization terms that incorporate prior information and mitigate invalid causality, addressing issues related to relaxed constraints. Inspired by the Generalized Jensen-Shannon Divergence, this paper contributes to establishing an upper bound for domain alignment in domain-independent conditional features. It reveals that many previous mDG approaches only partially optimize the objective, resulting in limited performance. Consequently, the general learning objective is distilled into four practical components, making it easily applicable across various tasks and different frameworks.

**Strengths:**

1.This paper provides a mathematical analysis of the multi-domain generalization problem, offering a unified learning objective.

2.The paper exhibits a clear logical structure with well-organized content.

3.The proposed method is straightforward and effective, capable of further enhancing existing approaches.

**Weaknesses:**

1.To offer a more comprehensive view of the proposed method, it would be beneficial for the authors to address its limitations.

2.While the ablation study is informative, it remains somewhat limited, as it focuses solely on verification using the "office-home" dataset. The authors should consider extending validation experiments to more datasets and tasks to robustly establish the method's effectiveness.

3.In the comparative analysis, the choice of methods for comparison is somewhat dated. It would be advisable for the authors to incorporate some recent domain generalization works published in 2023 for more up-to-date comparisons.

4.The method's description heavily relies on mathematical formulas, which may be less accessible to non-theoretical researchers. It is recommended to include textual explanations to clarify why the proposed method is effective.

5.The reference formatting presents some issues, such as missing "In" before "ECCV." It's essential to thoroughly review and correct these minor errors.

**Questions:**

Please see the weaknesses.

---

### Official Review · Reviewer_6BEZ · 2023-11-10

**Soundness:** 2 fair
**Presentation:** 2 fair
**Contribution:** 2 fair
**Rating:** 5
**Confidence:** 2

**Summary:**

This paper aims to solve the setting of multi-domain generalization (mDG). It provides a general learning objective paradigm, which proposes to leverage a $Y$-mapping to relax the constraint of a static target's marginal distribution. The general objective mainly has two aims: learning domain-independent conditional features and maximizing a posterior. Two regularization terms incorporating prior information and suppressing invalid causality are also involved. Based on the Generalized Jensen-Shannon Divergence, it also derives an upper bound for the domain-independent conditional features. The relationships between the four parts of the general objective and related work are also discussed. Finally, experimental results illustrate the effectiveness of the involvement of the $Y$-mapping in learning.

**Strengths:**

First of all, I must admit that I am not an expert in the field of multi-domain generalization, and may miss some related work.

## Originality
* The involvement of $Y$-mapping in the general learning objective seems novel.

## Quality
* It is good to discuss the relationships between the proposed general learning objective and existing methods.
* The experimental results illustrate the effectiveness of the proposed general learning objective.

## Clarity
*  Overall, this paper is written somewhat clearly.

## Significance
* This paper can contribute to the sub-area of multi-domain generalization.

**Weaknesses:**

## Originality
* The proposed general learning objective has a high similarity with the MDA method. Thus, the novelty of associated parts except the $Y$-mapping in the general learning objective is limited.

## Quality & Clarity
* Although this paper claims the theory contribution, I have not seen formal results and the theoretical contributions may be over-claimed.
* Experimentally, the computational cost of the proposed framework has not been discussed.
* The structure of this paper needs to be improved. For example, while the name of Section 4 is "Validating Proposed Objective: Theoretical Analysis", I have not seen some formal theoretical results but many experimental results.

## Significance
* This paper may have little effect on other sub-areas of machine learning.

**Questions:**

1. While the proposed general learning objective involves more terms, how to tune the hyper-parameters in practice?

2. Computationally, does the proposed method have a higher cost than related baselines?

3. In Table. 1, the part of (Ours, Reg1), why optimize the $\psi$ while the objective function has no function of it?

4. In Table. 2 (and Eq.(10)), it is weird to consider the mapping $C: \psi(Y) \rightarrow Y$ since we cannot do that in test time. Please give explanations.